# GPR30 in spinal cholecystokinin-positive neurons modulates neuropathic pain

Qing Chen[1,2,3†], Hui Wu[1,3†], Shulan Xie[1,2,3†], Fangfang Zhu[1,3], Fang Xu[4], Qi Xu[1,3], Lihong Sun[1,3], Yue Yang[5], Linghua Xie[1,3], Jiaqian Xie[6], Hua Li[1,3], Ange Dai[1,3], Wenxin Zhang[1], Luyang Wang[1,3], Cuicui Jiao[1,3], HongHai Zhang[5], Xuelong Zhou[1,3], Zhen-Zhong Xu[2,7,8]*, Xinzhong Chen[1,3]*

[1]Department of Anesthesia, Women's Hospital, Zhejiang University School of Medicine, Hangzhou, China; [2]NANHU Brain-Computer Interface Institute, Hangzhou, China; [3]Zhejiang Key Laboratory of Maternal and Infant Health, Hangzhou, China; [4]Department of Anesthesiology, The Second Affiliated Hospital, Chongqing Medical University, Chongqing, China; [5]Department of Anesthesiology, Affiliated Hangzhou First People's Hospital, Westlake University School of Medicine, Hangzhou, China; [6]Department of Anesthesiology, Sir Run Run Shaw Hospital, School of Medicine, Zhejiang University, Hangzhou, China; [7]Department of Anesthesiology, First Affiliated Hospital and School of Brain Science and Brain Medicine, Zhejiang University School of Medicine, Hangzhou, China; [8]Liangzhu Laboratory, MOE Frontier Science Center for Brain Science and Brain-machine Integration, State Key Laboratory of Brain-machine Intelligence, NHC and CAMS Key Laboratory of Medical Neurobiology, Zhejiang University, Hangzhou, China

*For correspondence:
xuzz@zju.edu.cn (Z-ZX);
chenxinz@zju.edu.cn (XC)

†These authors contributed equally to this work

Competing interest: The authors declare that no competing interests exist.

## eLife Assessment

This **important** study investigates nerve-injury-induced allodynia by studying the role of a subpopulation of excitatory dorsal horn CCK+ neurons that express the estrogen receptor GPR30 and potentially modulate nociceptive sensitivity via direct inputs from primary somatosensory cortex. In this revised version, the authors addressed many of the critiques raised through added analyses that **convincingly** support the notion that spinal GPR30 neurons are indeed an excitatory subpopulation of CCK+ neurons that contribute to neuropathic pain. While evidence of a direct functional corticospinal projection to CCK+/GPR30+ neurons is not fully demonstrated, this work will be of broad interest to researchers interested in the neural circuitry of pain.

**Abstract** Neuropathic pain, a major health problem affecting 7–10% of the global population, lacks effective treatment due to its elusive mechanisms. Cholecystokinin-positive (CCK⁺) neurons in the spinal dorsal horn (SDH) are critical for neuropathic pain, yet the underlying molecular mechanisms remain unclear. Here, we show that the membrane estrogen receptor G-protein coupled estrogen receptor (GPER/GPR30) in spinal neurons was significantly upregulated in chronic constriction injury (CCI) mice and that inhibition of GPR30 in CCK⁺ neurons reversed CCI-induced neuropathic pain. Furthermore, GPR30 in spinal CCK⁺ neurons was essential for the enhancement of AMPA-mediated excitatory synaptic transmission in CCI mice. Moreover, GPR30 was expressed in spinal CCK⁺ neurons that received direct projection from the primary sensory cortex (S1-SDH). Chemogenetic inhibition of S1-SDH post-synaptic neurons alleviated CCI-induced neuropathic pain. Conversely, chemogenetic activation of these neurons mimicked neuropathic pain symptoms, which were attenuated by spinal inhibition of GPR30. Finally, we confirmed that GPR30 in S1-SDH post-synaptic neurons was required for CCI-induced neuropathic pain. Taken together, our findings

suggest that GPR30 in spinal CCK[+] neurons and S1-SDH post-synaptic neurons is pivotal for neuropathic pain, thereby representing a promising therapeutic target for neuropathic pain.

## Introduction

Neuropathic pain, a persistent and disabling condition affecting approximately 7–10% of the global population, arises from a lesion or dysfunction of the somatosensory nervous system (*Ghazisaeidi et al., 2023*; *Xia et al., 2021*; *Cohen et al., 2021*; *Bannister et al., 2020*). This disorder manifests as mechanical allodynia (pain triggered by innocuous stimuli) and thermal hyperalgesia (exaggerated pain responses to noxious heat) (*Ghazisaeidi et al., 2023*; *Finnerup et al., 2021*; *Lolignier et al., 2015*). However, the complex and elusive mechanisms of neuropathic pain still make it difficult to treat. Therefore, identifying precise mechanisms and discovering new therapeutic targets is urgently needed.

The spinal cord (SC) serves as a pivotal hub for processing peripheral sensory inputs and integrating descending modulatory signals. Neuropathic mechanical allodynia occurs via circuit-based transformation in this region (*Bannister et al., 2020*; *Finnerup et al., 2021*; *Cheng et al., 2017*; *Ossipov et al., 2000*). Superficial laminae of the SC primarily encode noxious stimuli, whereas deeper laminae process innocuous signals. Under neuropathic conditions, however, low-threshold mechanosensory inputs aberrantly activate nociceptive neurons in superficial laminae through disinhibition and sensitization mechanisms, driving mechanical allodynia. Excitatory interneurons, constituting ~75% of spinal dorsal horn (SDH) neurons, are central to this pathological process (*Peirs and Seal, 2016*; *Peirs et al., 2021*; *Abraira et al., 2017*). Among these, cholecystokinin-expressing (CCK[+]) interneurons, enriched in SDH deep laminae, have recently been proposed as mediators of both mechanical and thermal hypersensitivity (*Peirs et al., 2021*; *Wang et al., 2022*). Notably, CCK[+] neurons receive direct corticospinal projections from the primary sensory cortex (S1), modulating neuropathic pain sensitivity (*Liu et al., 2018*); however, the molecular underpinnings of this regulation remain obscure.

Estrogen, beyond its classical nuclear receptors (ERα/ERβ), exerts non-genomic effects via the membrane receptor G protein-coupled estrogen receptor (GPR30), which is increasingly recognized as a key player in nociceptive modulation (*Li et al., 2023*; *Lee et al., 2023*; *Xu et al., 2022*; *Vannuccini et al., 2022*; *Sharp et al., 2022*; *Tramullas et al., 2021*; *Pang et al., 2023*; *Duan et al., 2021*; *Arnal et al., 2017*; *Chen et al., 2021*; *Jiao et al., 2023*; *Luo et al., 2016*; *Takanami et al., 2010*). Although GPR30's role in pain modulation has been documented, its specific contribution to SDH circuitry in neuropathic pain remains unexplored.

In this study, we demonstrate that spinal GPR30 orchestrates neuropathic pain by modulating excitability of CCK[+] neurons and corticospinal descending facilitation. We discovered that GPR30 activation in spinal CCK[+] neurons was both required and sufficient for the development of neuropathic pain. Interestingly, we observed that GPR30 in spinal CCK[+] neurons was required for the enhancement of spontaneous excitatory post-synaptic currents (sEPSC) in CCI mice, in an AMPA-dependent manner. Importantly, we revealed that GPR30 was expressed in the spinal CCK[+] neurons receiving direct projections from S1 sensory cortex, and that GPR30 in S1-SDH post-synaptic neurons was critical for CCI-induced neuropathic pain. These findings establish GPR30 in spinal CCK[+] neurons as a compelling therapeutic target for neuropathic pain management.

## Results

### Spinal inhibition of GPR30 reverses CCI-induced neuropathic pain and neuronal activation

To investigate the functional relevance of spinal GPR30 in neuropathic pain, we employed a chronic constriction injury (CCI) model to evaluate whether pharmacological blockade of spinal GPR30 alleviates neuropathic pain (*Figure 1A*). Quantitative PCR (qPCR) analysis revealed a significant elevation of *Gper1* (GPR30) mRNA levels in the lumbar SDH of CCI mice compared to sham (*Figure 1B*). In addition, the *Gper1* mRNA levels in the DRG remained unchanged after CCI (*Figure 1B*). Intrathecal application of the GPR30 antagonist, G-15, effectively attenuated CCI-induced mechanical allodynia and thermal hyperalgesia in both sexes of mice (*Figure 1C–E*), whereas basal nociceptive thresholds

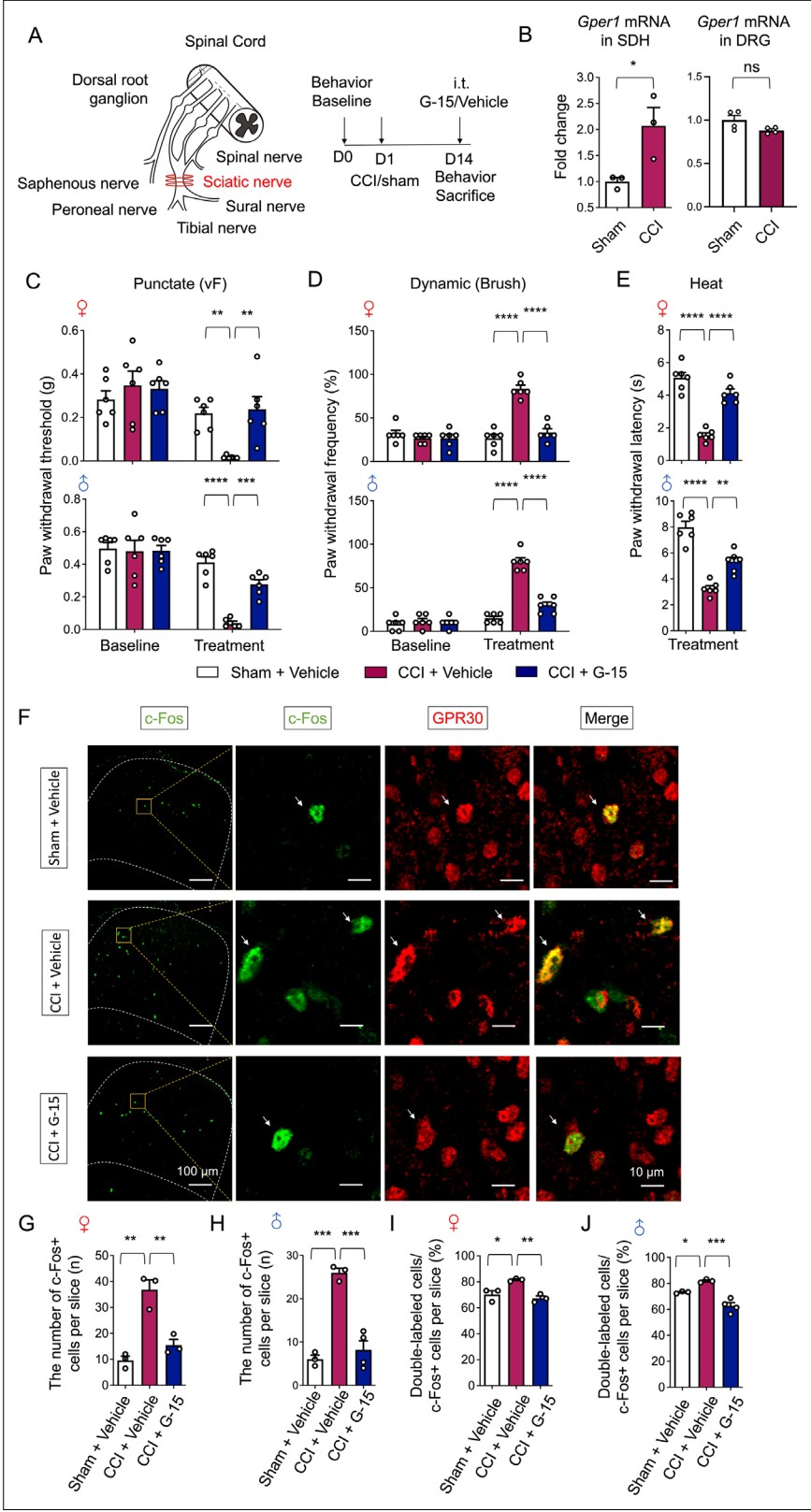

**Figure 1.** Chronic constriction injury (CCI)-induced neuropathic pain and neuronal activation were reversed by spinal inhibition of GPR30. (**A**) Schematic illustration of CCI surgery for induction of neuropathic pain (left) and diagram showing the timeline of CCI surgery, drug administration, and behavioral tests (right). (**B**) Quantitative PCR analysis of *Gper1* mRNA in spinal dorsal horn (SDH) from sham and CCI mice (left, n=3 mice for each group)

*Figure 1 continued*

and *Gper1* mRNA in DRG from sham and CCI mice (right, n=4 mice for each group). (**C–E**) Behavioral tests of basic nociception and 14 days after CCI or sham surgery along with intrathecal injection of GPR30 antagonist or vehicle in Von Frey tests (**C**), Brush tests (**D**), and Heat tests (**E**) in mice of both sexes (n=6 mice for each group). (**F**) Immunochemical detection of c-Fos (green) and GPR30 (red). Scale bars: 100 μm. Boxed area of images is enlarged on the right. Scale bars: 10 μm. White arrows indicate double-positive cells. (**G, H**) Total number of c-Fos-positive neurons in the SDH per section in female mice (**G**) and male mice (**H**) (n=3–4 mice for each group, 4–6 pictures were analyzed for each mouse). (**I, J**) Percentage of c-Fos-positive neurons expressing GPR30 in female mice (**I**) and male mice (**J**) (n=3–4 mice for each group, 4–6 pictures were analyzed for each mouse). Data information: in (**B**), *p<0.05 (unpaired Student's *t*-test). In (**C, D**), **p<0.01; ***p<0.001; ****p<0.0001 (two-way ANOVA with Turkey's multiple comparisons test). In (**E**), **p<0.01; ****p<0.0001 (one-way ANOVA with Turkey's multiple comparisons test). In (**G–J**), *p<0.05; **p<0.01; ***p<0.001 (one-way ANOVA with Turkey's multiple comparisons test). All data are presented as mean ± SEM.

The online version of this article includes the following source data and figure supplement(s) for figure 1:

**Source data 1.** Numerical data and statistic methods used in *Figure 1* and *Figure 1—figure supplement 1*.

**Figure supplement 1.** Spinal inhibition of GPR30 did not change the basic nociception while spinal activation of GPR30 mimicked neuropathic pain symptoms in naïve mice.

remained unaltered in naïve mice (*Figure 1—figure supplement 1A and B*). Of note, the analgesic effects of G-15 lasted for up to 6 hours after intrathecal application in CCI mice (*Figure 1—figure supplement 1C*). Immunochemical data demonstrated that innocuous tactile stimulation triggered pronounced c-Fos expression, a marker of neuronal activation, predominantly in GPR30⁺ cells within the SDH of CCI mice (*Figure 1F–J*). Critically, G-15 treatment substantially suppressed this c-Fos expression (*Figure 1F–J*), indicating that spinal GPR30 inhibition normalizes both hypersensitivity and neuronal activation in neuropathic pain in both sexes.

We further probed whether GPR30 activation alone could mimic neuropathic symptoms in naïve mice. Intrathecal application of the selective GPR30 agonist, G-1, robustly evoked mechanical allodynia and thermal hypersensitivity, with symptom persistence exceeding 48 hours (*Figure 1—figure supplement 1D–G*). Consistent with behavioral outcomes, G-1 administration markedly increased c-Fos⁺ neurons in the SDH, with over 80% of activated cells co-expressing GPR30 (*Figure 1—figure supplement 1H–J*).

Collectively, these findings establish spinal GPR30 as a critical mediator of sex-independent neuropathic pain development in the CCI model.

## GPR30 in spinal CCK⁺ neurons is required for CCI-induced neuropathic pain

To delineate the spinal cellular substrates of GPR30-mediated pain modulation, we first mapped its expression pattern. Our results revealed exclusive GPR30 localization to neuronal populations within the SDH instead of astrocytes or microglial cells via immunochemistry. Notably, as an estrogen receptor, GPR30 expression showed no sexual dimorphism (*Figure 2—figure supplement 1*). To assess the relative expression of GPR30 in excitatory versus inhibitory neurons, we performed in situ hybridization of *Slc32a1* and *Slc17a6*, combined with immunostaining of GPR30 (*Figure 2A*). GPR30 was predominantly expressed on the excitatory neurons of SDH (*Figure 2B*). Further characterization in wild-type mice using AAV2/9-Camk2-mCherry labeling also confirmed predominant GPR30 enrichment in SDH excitatory interneurons (*Figure 2—figure supplement 2*).

Given the established role of CCK⁺ neurons in neuropathic pain and their dominance among SDH excitatory interneurons (*Peirs et al., 2021*; *Duan et al., 2021*; *Qi et al., 2023*; *Koch et al., 2018*), we hypothesized that GPR30 within this subset drives nociceptive sensitization. To validate this, we characterized CCK expression in the SC via transgenic models by crossing *Cck^Cre* with *Rosa26^tdTomato* reporter mice (Ai14) (*Figure 2C*). In situ hybridization confirmed that our transgenic mice recapitulated the majority of *Cck* expression (*Figure 2D*), and immunostaining further showed CCK⁺ neurons predominantly distributed in the deep laminae of the SDH as marked by isolectin B4 (IB4), CGRP, and NF200 (*Figure 2E*). To verify the role of CCK⁺ neurons in allodynia under neuropathic pain, we performed immunostaining of c-Fos in CCK-Ai14 mice, and we confirmed that more CCK⁺ neurons

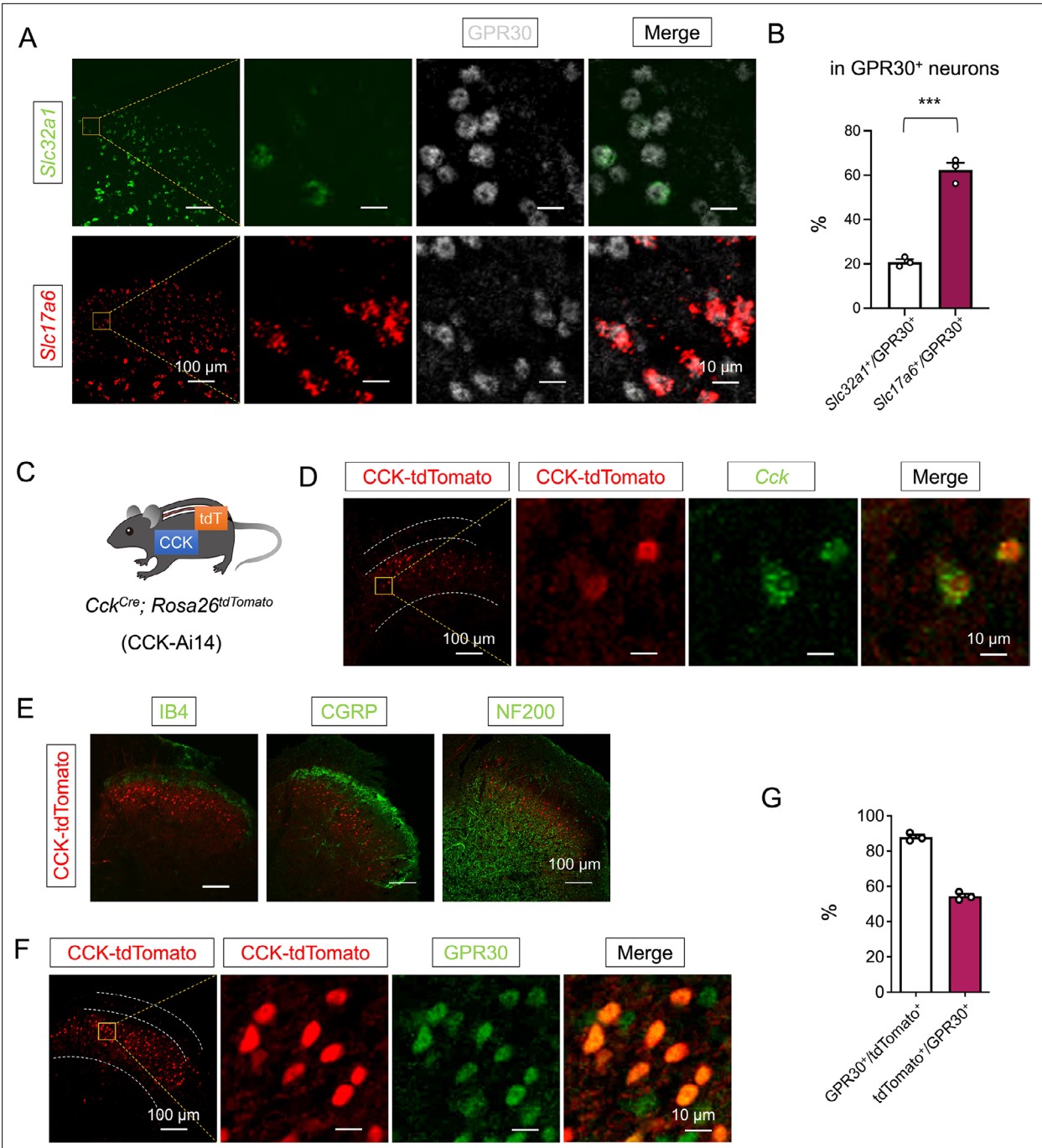

**Figure 2.** GPR30 was widely expressed in the CCK⁺ excitatory interneurons in the spinal dorsal horn (SDH). (**A**) In situ hybridization showing *Slc32a1* (green) and *Slc17a6* (red) with GPR30 (gray) in the SDH. Scale bar: 100 μm. Boxed area of the image is enlarged on the right. Scale bars: 10 μm. (**B**) Percentage of GPR30-positive neurons expressing *Slc32a1* or *Slc17a6* in wild mice (n=3 mice, 3–4 pictures were analyzed for each mouse). (**C**) *Cck^Cre* mice express tdTomato in a Cre-dependent manner. (**D**) In situ hybridization showing *Cck* (green) with tdTomato (red) in the SDH. Scale bar: 100 μm. Boxed area of image is enlarged on the right. Scale bars: 10 μm. (**E**) Double staining of tdTomato⁺ neurons (red) with IB4, CGRP, or NF200 (green) by immunochemistry. Scale bars: 100 μm. (**F**) Immunofluorescence in SDH to verify the co-localization of tdTomato⁺ neurons (red) with GPR30 (green). Scale bars: 100 μm. Boxed area of images is enlarged on the right. Scale bars: 10 μm. (**G**) Percentage of double-labeled neurons in CCK-tdTomato and GPR30-positive neurons (n=3 mice, three pictures were analyzed for each mouse) Data information: in (**B**), ***p<0.001 (unpaired Student's t-test). All data are presented as mean ± SEM.

The online version of this article includes the following source data and figure supplement(s) for figure 2:

**Source data 1.** Numerical data and statistic methods used in *Figure 2* and *Figure 2—figure supplements 1–3*.

*Figure 2 continued on next page*

*Figure 2 continued*

**Figure supplement 1.** GPR30 was mainly expressed in the spinal neurons.

**Figure supplement 2.** GPR30 was widely expressed in the Camk2+ excitatory interneurons in the spinal dorsal horn (SDH).

**Figure supplement 3.** CCK-positive neurons were more activated after chronic constriction injury (CCI).

were activated after CCI (*Figure 2—figure supplement 3*). Moreover, near-universal CCK+ neurons were co-localized with GPR30 (*Figure 2F and G*).

To functionally interrogate GPR30 and CCK+ neurons in neuropathic pain, we injected AAV2/9-DIO-shGper1-EGFP into lumbar SDH of *Cck^Cre* mice for conditional knock-down of *Gper1*, and 3 weeks later we subjected mice to CCI as well as subsequent behavioral tests (*Figure 3A*). Viral targeting accuracy was verified by immunostaining showing deep laminae-specific expression, mirroring endogenous CCK+ neurons distribution (*Figure 3B*). CCI-induced *Gper1* mRNA upregulation was significantly blunted in knockdown mice (*Figure 3C*, *Figure 4—figure supplement 1A*), confirming the transcriptional efficacy of virus.

Behavioral tests revealed that *Gper1* knockdown in CCK+ neurons attenuated CCI-induced mechanical allodynia and thermal hyperalgesia across sexes, without altering basal nociceptive thresholds (*Figure 3D–F*, *Videos 1 and 2*). To dissect affective components of neuropathic pain, we used real-time place escape/avoidance (RT-PEA) test with innocuous mechanical stimulation as shown before (*Xia et al., 2021*; *Figure 3G*). Scramble-treated CCI mice exhibited persistent aversion to the stimulated side, whereas *Gper1* knockdown abolished this aversive behavior during the post-stimulation period (*Figure 3H–L*, *Videos 3–8*).

## GPR30 in spinal CCK+ neurons is essential for the enhancement of excitatory synaptic transmission in the SDH of CCI mice

To investigate the role of GPR30 expressed on CCK+ neurons in neuropathic pain transmission, we conducted electrophysiological recordings using whole-cell patch clamp recording on fluorescently labeled neurons in spinal slices from CCI or sham-operated *Cck^Cre* mice treated with shRNA (*Figure 4A and B*). Since CCK+ neurons mainly receive synaptic inputs from upstream neurons, we then intended to test whether GPR30 modulated these synaptic connections. We recorded post-synaptic currents from GFP-labeled CCK+ neurons in laminae III–V approximately 2 weeks after CCI surgery. As hypothesized, the amplitude of spontaneous excitatory post-synaptic currents (sEPSCs) in CCK+ neurons was significantly enhanced in CCI mice compared to sham-operated mice, although the frequency remained unchanged (*Figure 4C–E*). Notably, this enhancement in amplitude was abolished in *Gper1* knockdown mice (*Figure 4C and E*). Notably, spontaneous inhibitory post-synaptic currents (sIPSCs) also contribute to the regulation of neuronal excitability. However, our findings indicated that knockdown of *Gper1* did not affect the IPSCs of spinal CCK+ neurons in CCI mice (*Figure 4—figure supplement 1*).

Given that EPSCs are primarily mediated through glutamatergic receptors such as AMPA receptor, coupled with emerging evidence showing that GPR30 could facilitate excitatory transmission via increasing the clustering of glutamatergic receptor subunits (*Luo et al., 2016*), we proceeded to examine whether GPR30 activation modulated AMPA-mediated currents. Electrophysiological experiments were carried out in spinal slices from *Cck^Cre* mice treated with intraspinal injection of AAV2/9-DIO-EGFP (*Figure 4F*). To record AMPA-dependent currents, electronic stimulation from the electrode placed in the deep laminae of SDH was applied (*Figure 4G*). The voltage of CCK+ neurons was held at –70 mV in the presence of APV (100 µM) to block NMDA receptors and bicuculline (20 µM) to block GABA_A receptors and strychnine (0.5 µM) to block glycine receptors. All recorded cells responded to the electrode stimulation, exhibiting evoked EPSCs (*Figure 4H and I*). Furthermore, the application of 0.1 µM G-1 augmented the amplitude of AMPA currents (*Figure 4H and I*). These findings collectively suggest that GPR30 regulates sEPSCs in CCK+ neurons through an AMPA-dependent mechanism.

## GPR30 is expressed in the spinal CCK+ neurons receiving direct projection from S1 cortex

To anatomically verify that the lumbar SC receives direct descending projections from the S1 cortex, we employed a combination of retrograde and anterograde tracing methods to investigate the

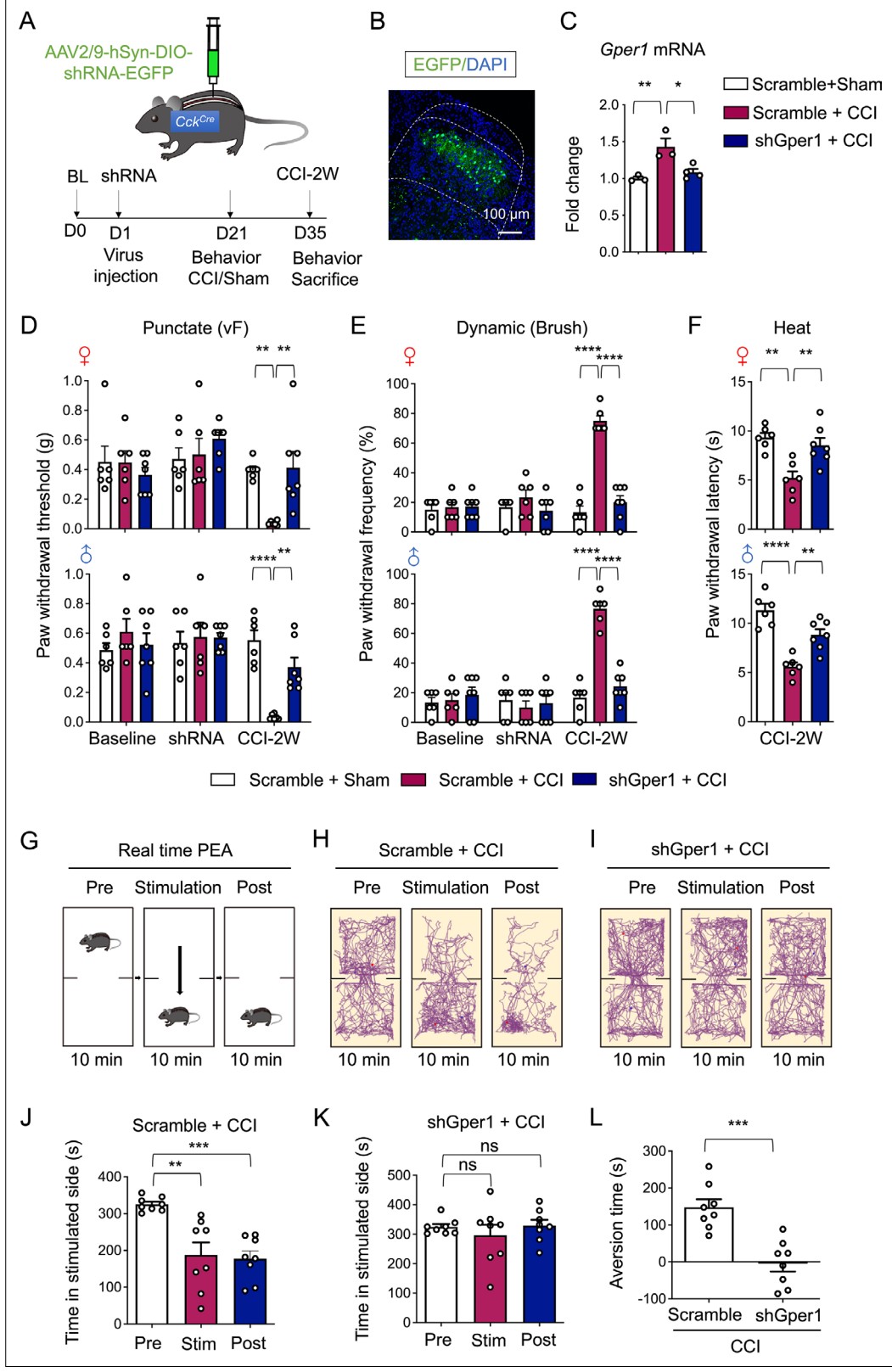

**Figure 3.** Knockdown of GPR30 in spinal CCK[+] neurons alleviated neuropathic pain in chronic constriction injury (CCI) mice. (**A**) Schematic illustration of the strategy to knock down Gper1 in the spinal CCK[+] neurons in a Cre-dependent manner (up) and the diagram showing the timeline of virus injection, CCI surgery, and behavioral tests (down). (**B**) Immunochemical detection of the localization of virus expression (green). Scale bars: 100 μm.

*Figure 3 continued on next page*

*Figure 3 continued*

(**C**) Quantitative PCR analysis of Gper1 mRNA in spinal dorsal horn (SDH) from sham and CCI mice with intraspinal virus injection (n=3–4 mice for each group). (**D–F**) Behavioral tests of basic nociception, 21 days after spinal virus injection and 14 days after CCI or sham surgery along with intrathecal injection of antagonist of GPR30 or vehicle in Von Frey tests (**D**), Brush tests (**E**), and Heat tests (**F**) in mice of both sexes (n=6–7 mice for each group). (**G**) Schematic illustration of real-time place escape/avoidance (PEA) test. (**H, I**) Representative spatial tracking maps showing the location of a Scramble + CCI group mice (**H**) and a shGper1+CCI group mice (**I**) before, during, and after mechanical stimulation to the hind paw in the chambers. (**J**) Quantification of time the Scramble + CCI group mice spent in preferred chamber before, during, and after stimulation (n=8 mice). (**K**) Quantification of time the shGper1+CCI group mice spent in preferred chamber before, during, and after stimulation (n=8 mice). (**L**) Quantification of the aversion time of mice (time spent in preferred chamber after stimulation minus the time spent before stimulation) (n=8 mice for each group). Data information: in (**C**), $*p<0.05$; $**p<0.01$ (one-way ANOVA with Turkey's multiple comparisons test). In (**D, E**), $**p<0.01$; $****p<0.0001$ (two-way ANOVA with Turkey's multiple comparisons test). In (**F**), $**p<0.01$; $****p<0.0001$ (one-way ANOVA with Turkey's multiple comparisons test). In (**J, K**), $*p<0.05$; $**p<0.01$; $***p<0.001$; ns = not significant. (one-way ANOVA with Turkey's multiple comparisons test). In (**L**), $***p<0.001$ (unpaired Student's t-test). All data are presented as mean ± SEM.

The online version of this article includes the following source data for figure 3:

**Source data 1.** Numerical data and statistic methods used in *Figure 3*.

connections between cortical neurons and their axonal innervation in the SDH. Our results revealed that retrograde labeling with cholera toxin subunit B-555 (CTB-555), injected into the deep laminae of the lumbar SDH, predominantly localized neurons in the contralateral S1 cortex (*Figure 5A–C*). Of note, the neurons projecting from S1 cortex are primarily excitatory pyramidal neurons located in layer V (*Cai et al., 2023*). Similarly, fiber tracing using AAV2/9-hSyn-EGFP indicated that axons from S1 terminate extensively in laminae III–V of the contralateral SDH (*Figure 5D–F*).

Building on previous findings suggesting a functional interaction between S1-SDH projections and spinal CCK[+] neurons (*Liu et al., 2018*), our current study aimed to further elucidate the structural relationship among GPR30, S1-SDH projections, and CCK[+] neurons. To achieve this, we performed anterograde trans-monosynaptic tracing by injecting AAV2/1-hSyn-Cre into the right S1 cortex, followed by AAV2/9-DIO-mCherry into the contralateral lumbar SDH. This approach allowed us to visualize the mCherry[+] S1-SDH post-synaptic neurons in the deep laminae of the lumbar SDH (*Figure 5G and H*). Consistent with prior observations, we found that 28.1% of mCherry+S1-SDH downstream neurons coexpressed *Cck* (*Figure 5I and J*). Thus, our findings confirm that spinal CCK[+] neurons are innervated by the S1 cortex.

To further examine the interplay between GPR30, S1-SDH projections, and CCK[+] neurons, we utilized a monosynaptic anterograde tracing strategy. Specifically, we injected AAV2/1-EF1α-FLP into the right S1 cortex and AAV2/9-hSyn-Con/Fon-GFP into the contralateral lumbar SDH of *Cck^Cre* mice. This strategy enabled us to visualize GFP-positive S1-SDH post-synaptic CCK[+] neurons (*Figure 5K and L*). Our co-staining results revealed that the vast majority of CCK[+] S1-SDH post-synaptic neurons

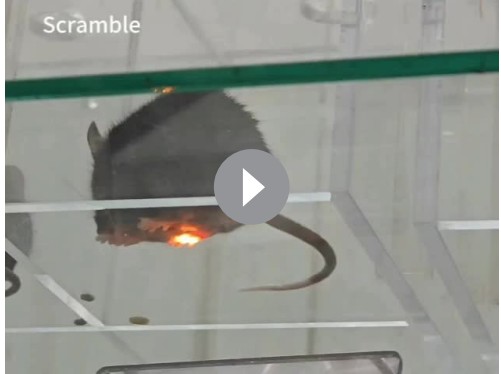

**Video 1.** Heat test of mice subjected to chronic constriction injury and spinal viral injection.
https://elifesciences.org/articles/102874/figures#video1

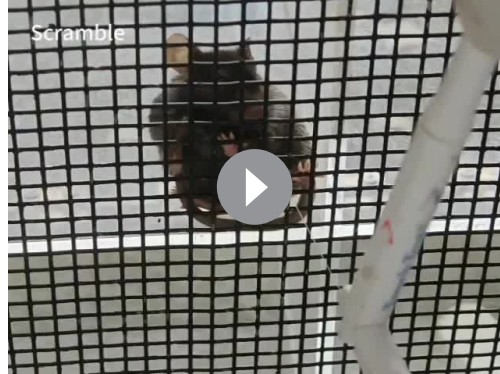

**Video 2.** Von Frey test of mice subjected to chronic constriction injury and spinal viral injection.
https://elifesciences.org/articles/102874/figures#video2

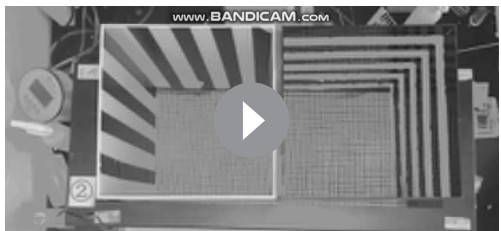

**Video 3.** Pre-stimulation phase of real-time place escape/avoidance test of mice subjected to chronic constriction injury and shGper1.
https://elifesciences.org/articles/102874/figures#video3

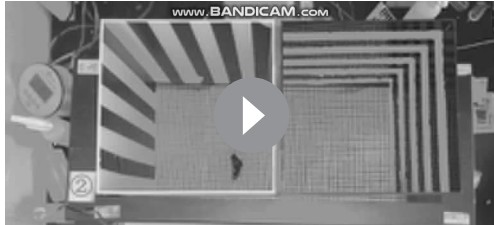

**Video 4.** Stimulation phase of real-time place escape/avoidance test of mice subjected to chronic constriction injury and shGper1.
https://elifesciences.org/articles/102874/figures#video4

expressed GPR30 (*Figure 5M and N*). These data collectively indicate that the majority of CCK⁺ neurons receiving S1 projections express GPR30.

## GPR30 in S1-SDH post-synaptic neurons is critical for CCI-induced neuropathic pain

Given that GPR30 has been verified to be expressed on CCK⁺ neurons receiving S1-SDH direct projections (*Figure 5M*), we employed a combination of chemogenetic and pharmacological approaches to determine whether neurons innervated by S1-SDH direct projections mediate nociception via GPR30 (*Wu et al., 2021*). Specifically, we performed injections of anterograde AAV2/1-hSyn-Cre into the right S1 WT mice. One week later, we administered AAV2/9-hSyn-DIO-hM3Dq (Gq)-mCherry into the lumbar SDH (*Figure 6A*). Pharmacological activation of these post-synaptic neurons with CNO in Gq-treated mice significantly induced spontaneous pain-like behaviors, such as paw scratching, biting, and licking, in the hind paws and tails (*Figure 6B*). Furthermore, chemogenetic activation of S1-SDH post-synaptic neurons dramatically induced mechanical allodynia and thermal hyperalgesia in both sexes compared to negative control groups (*Figure 6C–E*). Notably, the reduction in nociceptive thresholds could be effectively reversed by intrathecal administration of G-15 (*Figure 6C–E*). The corresponding immunohistochemistry results using c-Fos confirmed the chemogenetic activation of S1-SDH post-synaptic neurons, which could be suppressed by intrathecal application of G-15 (*Figure 6F–H*).

To further explore the role of S1-SDH post-synaptic neurons in the modulation of neuropathic pain, we employed chemogenetic inhibitory methods to suppress these neurons in CCI mice. The suppression of S1-SDH post-synaptic neurons could dramatically relieve the mechanical allodynia and thermal hyperalgesia induced by CCI (*Figure 6—figure supplement 1*). To establish the essential role of GPR30 in this process, we specifically knocked down the expression of *Gper1* on S1-SDH post-synaptic neurons and subjected mice to CCI after adequate viral expression (*Figure 7A*). Interestingly, the knockdown of *Gper1* in S1-SDH post-synaptic neurons was sufficient to relieve mechanical allodynia and thermal hyperalgesia in both sexes (*Figure 7B–D*). Immunochemistry showed the viral location in deep laminae of SDH (*Figure 7E*) and qPCR confirmed the suppression of *Gper1*

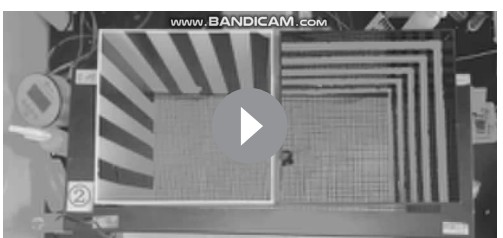

**Video 5.** Post-stimulation phase of RT-PEA test of mice subjected to chronic constriction injury and shGper1.
https://elifesciences.org/articles/102874/figures#video5

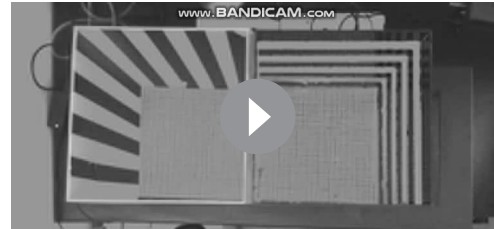

**Video 6.** Pre-stimulation phase of real-time place escape/avoidance test of mice subjected to chronic constriction injury and Scramble.
https://elifesciences.org/articles/102874/figures#video6

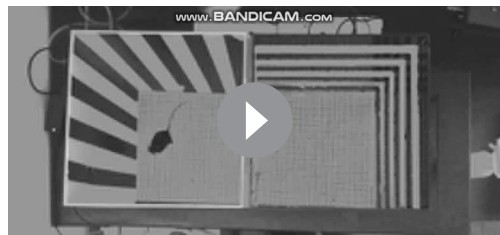

**Video 7.** Stimulation phase of real-time place escape/avoidance test of mice subjected to chronic constriction injury and Scramble.
https://elifesciences.org/articles/102874/figures#video7

mRNA expression which was increased by CCI (*Figure 7F*). Collectively, these findings underscore the role of GPR30 in the descending facilitation potentially mediated by S1-SDH projections in the neuropathic pain.

## Discussion

CCK[+] neurons, located in the deep laminae of the SC, have long been recognized for their pivotal role in the development and maintenance of neuropathic pain, as well as descending facilitation by sensory cortex-SC projections. Despite extensive research, the molecular mechanisms underlying nociception remain poorly understood. GPR30, as a membrane estrogen receptor, exerts modulatory effects on various physiological and pathological processes, including the development of neuropathic pain, within seconds to minutes (*Arnal et al., 2017*). In this study, we observed significant upregulation of GPR30 expression in the SDH after CCI, underscoring its critical role in neuropathic pain development. To investigate this further, we employed a comprehensive approach, including transgenic mice models, behavioral assays, pharmacological interventions, chemogenetic methods, and electrophysiological studies. These experiments revealed that GPR30 plays an indispensable role in neuropathic pain, particularly through its regulation of AMPA-dependent EPSCs in spinal CCK[+] neurons, which are essential for pain signaling. Furthermore, we demonstrated that GPR30 expressed on post-synaptic neurons of corticospinal direct projections took a role in modulation of neuropathic pain. These findings collectively suggest that GPR30 in spinal CCK[+] neurons may represent a promising therapeutic target for neuropathic pain.

Estrogen has long been recognized as a key player in modulating nociception, with its drastic fluctuations notably impacting nociceptive thresholds (*Zhang et al., 2020*). Our prior research demonstrated that moderate estrogen supplementation effectively mitigated hyperalgesia in ovariectomized (OVX) mice; however, excessive estrogen supplementation paradoxically exacerbated hyperalgesia (*Zhang et al., 2020*). This discrepancy might stem from the fact that varying estrogen concentrations can differentially activate estrogen receptors, including nuclear receptors (ERα and ERβ) and membrane receptors (GPR30) (*Arnal et al., 2017*). Considering our findings that inhibiting spinal GPR30 does not alter the basal nociception in naïve mice, it appears that GPR30 may not be significantly activated by estrogen under normal conditions. Additionally, estrogen has long been implicated in modulating neuropathic pain. Intra-dorsal root ganglion (DRG) administration of estrogen in CCI rats has been shown to enhance mechanical and thermal pain in an ERα-dependent manner (*Deng et al., 2017*). Conversely, estrogen supplementation in the anterior cingulate cortex (ACC) of CCI mice significantly alleviated neuropathic hyperalgesia in a GPR30-dependent manner (*Wang et al., 2024*). These findings further underscore the distinct roles of different estrogen receptors in pain modulation.

GPR30 is widely expressed in the nervous system and exerts vital effects in nociceptive modulation (*Xu et al., 2022*; *Chen et al., 2021*; *Jiao et al., 2023*; *Wang et al., 2024*; *Li et al., 2024*; *Jiang et al., 2020*; *Liu et al., 2015*; *Tian et al., 2013*). For example, the activation of GPR30 in DRG could aggravate the hyperalgesia in OVX mice, while inhibition of GPR30 relieved hyperalgesia (*An et al., 2014*). Besides, the GPR30 expressed on GABAergic cells in rostral ventromedial medulla (RVM) mediates the descending facilitation of nociception (*Jiao et al., 2023*). However, though GPR30 is also widely expressed in SDH (*Takanami et al., 2010*), still little is known about the functions as well as underlying mechanisms of spinal GPR30 in nociceptive modulation. Consistent with previous studies (*Deliu et al., 2012*), here we

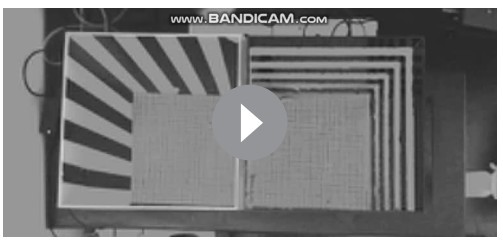

**Video 8.** Post-stimulation phase of real-time place escape/avoidance test of mice subjected to chronic constriction injury and Scramble.
https://elifesciences.org/articles/102874/figures#video8

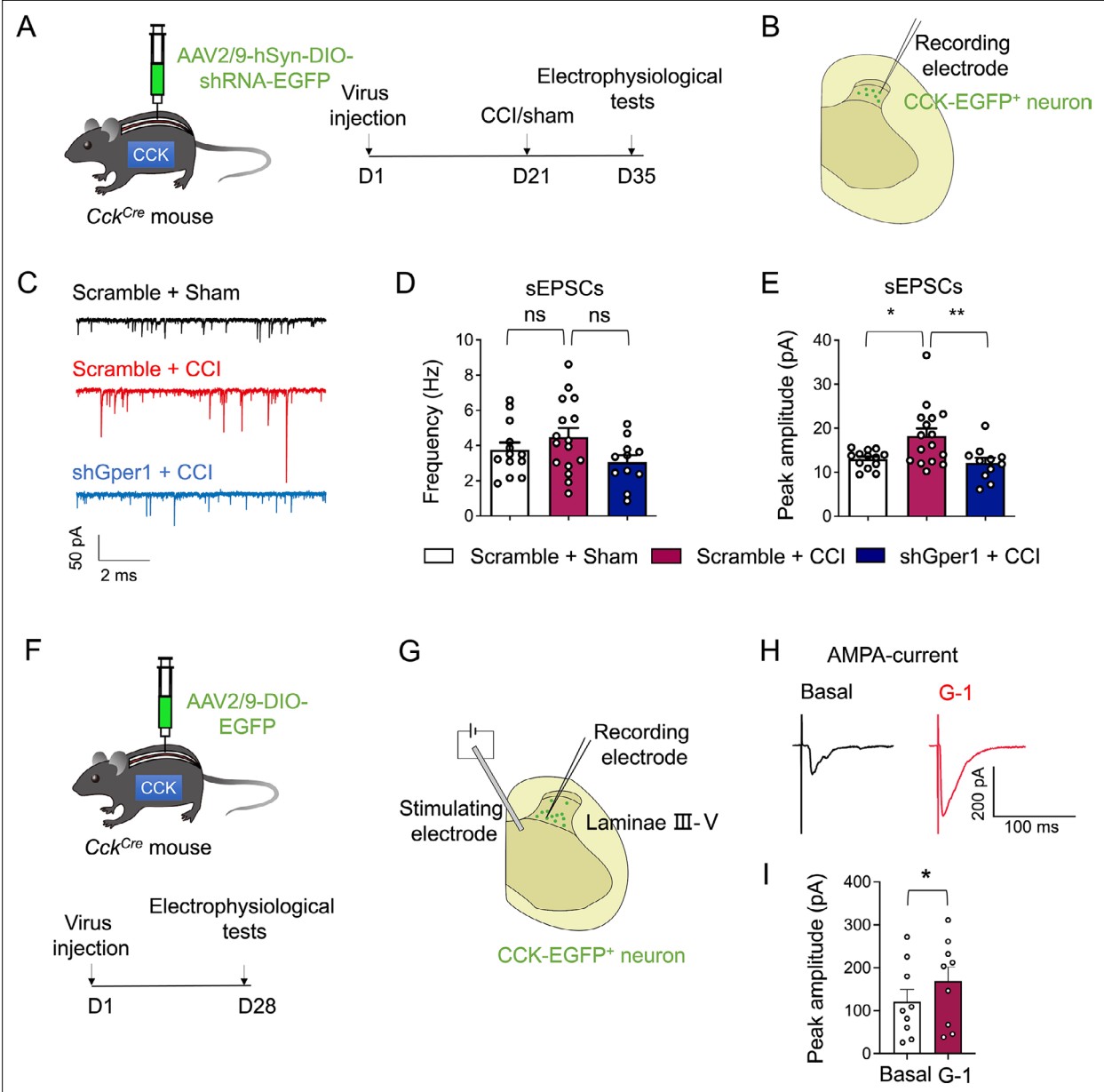

**Figure 4.** Knockdown of GPR30 in spinal CCK+ neurons reverses the enhancement of spontaneous excitatory post-synaptic currents (sEPSC) in chronic constriction injury (CCI) mice. (**A**) Schematic illustration of the strategy to knock down Gper1 in the spinal CCK+ neurons in a Cre-dependent manner (left) and the diagram showing the timeline of virus injection, CCI surgery, and electrophysiological tests (right). (**B**) Schematic illustration of spinal slice electrophysiological recordings from CCK-EGFP neurons located in the deep laminae of spinal dorsal horn (SDH). (**C**) Representative traces of sEPSCs in CCK-EGFP-positive neurons. (**D**) Quantification of sEPSCs frequency in CCK-EGFP-positive neurons (n=13–16 cells from five mice per group). (**E**) Quantification of sEPSCs amplitude in CCK-EGFP-positive neurons (n=13–16 cells from five mice per group). (**F**) Schematic illustration of the strategy to visualize CCK+ neurons in a Cre-dependent manner (up) and the diagram showing the timeline of virus injection and electrophysiological tests (down). (**G**) Schematic illustration of spinal slice electrophysiological recordings from CCK-EGFP+ neurons located in the deep laminae of SDH during electric stimulation of the deep laminae of SDH to simulate APMA currents. (**H**) Representative AMPA current in CCK-EGFP+ neurons before and after administration of G-1 (0.1 μM). (**I**) The peak amplitude of AMPA currents before and after administration of G-1 (n=8 cells from four mice). Data information: in (**D, E**), *p<0.05; **p<0.01; ns = not significant (one-way ANOVA with Turkey's multiple comparisons test). In (**I**), *p<0.05 (paired Student's t-test). All data are presented as mean ± SEM.

The online version of this article includes the following source data and figure supplement(s) for figure 4:

**Source data 1.** Numerical data and statistic methods used in *Figure 4* and *Figure 4—figure supplement 1*.

**Figure supplement 1.** Knockdown of GPR30 in spinal CCK+ neurons did not change the spontaneous inhibitory post-synaptic currents (sIPSCs) in chronic constriction injury (CCI) mice.

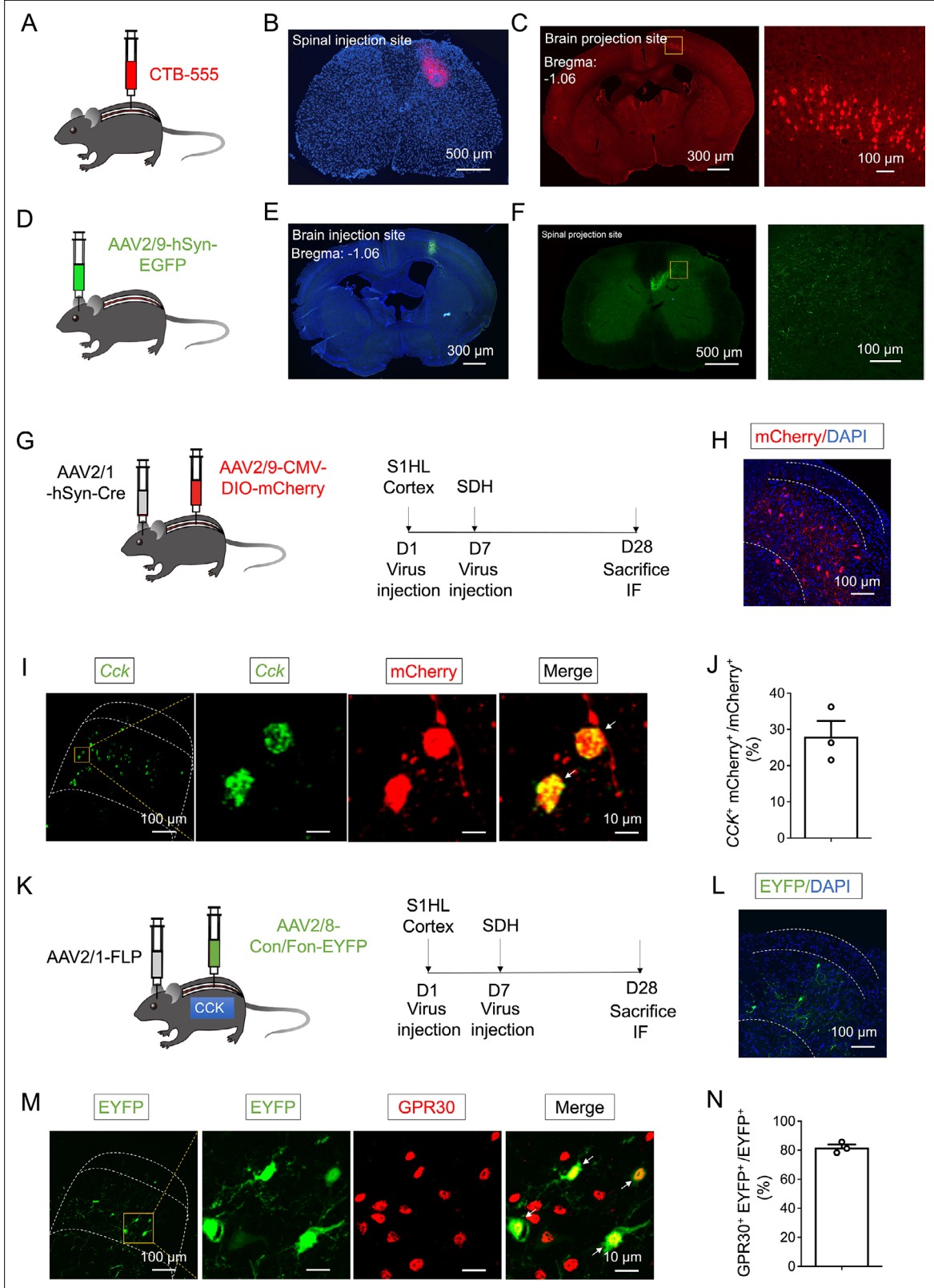

**Figure 5.** GPR30 was expressed in the spinal CCK+ neurons receiving direct projection from S1 cortex. (**A**) Schematic illustration of the strategy to retrograde tracing the projections innervating spinal dorsal horn via CTB-555. (**B**) Fluorescence image showing the site of CTB-555 injection in the spinal dorsal horn. Scale bars: 500 μm. (**C**) Coronal brain section showing the location of tdTomato + neurons in the S1HL cortex. Scale bars: 300 μm. Boxed area of image is enlarged on the right. Scale bars: 100 μm. (**D**) Schematic illustration of the strategy to anterograde tracing the projections from S1HL

*Figure 5 continued on next page*

*Figure 5 continued*

cortex. (**E**) Coronal brain section showing the site of AAV injection in the S1HL cortex. Scale bars: 300 μm. (**F**) Fluorescence image showing the S1HL-SDH tract terminates in the deep laminae of spinal dorsal horn. Scale bars: 500 μm. Boxed area of image is enlarged on the right. Scale bars: 100 μm. (**G**) Schematic illustration of the strategy for identifying the spinal post-synaptic neurons of S1-SDH projections (left) and diagram showing the timeline of AAV2/1 injection in the S1HL cortex, AAV2/9 injection in the spinal dorsal horn and immunofluorescence (right). (**H**) Representative image showing the mCherry⁺ post-synaptic neurons of the S1HL-SDH projections in the deep laminae of SDH. Scale bars: 100 μm. (**I**) In situ hybridization showing *Cck* (green) with RFP (red) by immunofluorescence in the spinal dorsal horn. Scale bars: 100 μm. Boxed area of image is enlarged on the right. Scale bars: 10 μm. White arrows indicate double-positive cells. (**J**) About 28.13% of RFP⁺ post-synaptic neurons of S1HL-SDH projections expressing CCK (n=3 mice, three pictures were analyzed for each mouse). (**K**) Schematic illustration of the strategy for identifying the spinal CCK⁺ post-synaptic neurons of S1HL-SDH projections in *Cck^Cre* mice (left) and diagram showing the timeline of AAV2/1 injection in the S1HL cortex, AAV2/9 injection in the spinal dorsal horn and immunofluorescence (right). (**L**) Representative image showing the EGFP⁺ CCK⁺ post-synaptic neurons of the S1-SDH projections in the spinal dorsal horn with DAPI. Scale bars: 100 μm. (**M**) Double staining of EGFP⁺ neurons with GPR30 by immunohistochemistry. Scale bars: 100 μm. Boxed area of image is enlarged on the right. Scale bars: 10 μm. (**N**) About 81.7% of CCK⁺ post-synaptic neurons of S1HL-SDH projections expressing GPR30 (n=3 mice, 7–18 pictures were analyzed for each mouse). Data information: in (**J**, **N**), data are presented as mean ± SEM.

The online version of this article includes the following source data for figure 5:

**Source data 1.** Numerical data and statistic methods used in *Figure 5*.

found that intrathecal injection of G-1 could dramatically induce mechanical allodynia and thermal hyperalgesia in mice. To further explore whether spinal GPR30 is involved in pathological nociception, we subjected mice to CCI surgery to mimic neuropathic pain (*Xia et al., 2021*; *Xiang et al., 2019*). In accordance with our expectations, the inhibition of spinal GPR30 significantly reversed the mechanical allodynia and thermal hyperalgesia induced by CCI. Moreover, GPR30 has been reported to be involved in nociceptive sexual dimorphism. For example, the regulatory role of GPR30 in DRG in maintenance of hyperalgesia induced by repeated exposure to opioid only exists in female rats (*Vacca et al., 2014*; *Vacca et al., 2016*). However, according to our results, spinal GPR30 modulated nociception in a sex-independent manner. We also found that GPR30 is indiscriminately expressed on the neurons of both sexes of mice, which might account for the sex-independent function of spinal GPR30 in nociceptive modulation. Consistent with our results, several studies also have confirmed that the spinal *Gper1* expression showed no significant difference between male and female mice (*Zhang et al., 2012a*). Furthermore, the fluctuation of estrogen failed to change the basic expression of spinal GPR30 (*Zhang et al., 2020*). These results further indicate that spinal GPR30 modulated nociception in a sex-independent manner.

The SDH is a major locus for the integration of peripheral sensory input and supraspinal modulation. Most peripheral nociceptive afferents project to the superficial laminae of the SDH which respond to the noxious stimulations, while low-threshold mechanoreceptors form synaptic contacts in the deep laminae which respond to innocuous stimulations (*Ossipov et al., 2000*). However, under the condition of mechanical allodynia, innocuous stimulation might also activate more superficial nociceptive circuits and lead to painful perception, which might come from the circuit-based transformation in the SDH (*Lolignier et al., 2015*; *Murthy et al., 2018*). It should be noted that the SDH is composed of a large number of excitatory (75%) and inhibitory (25%) interneurons, as well as a small part of projection neurons which relay integrated information to various supraspinal regions (*Peirs and Seal, 2016*). Excitatory interneurons have been confirmed to take a vital role in conveying mechanical allodynia according to the nature of injury (*Peirs et al., 2021*; *Arcourt and Lechner, 2015*). As a distinct type of excitatory interneurons mainly located in the deeper laminae of SDH, CCK⁺ neurons are important for neuropathic injuries (*Peirs et al., 2021*; *Abraira et al., 2017*; *Wang et al., 2022*). The inhibition of spinal CCK⁺ neurons could alleviate neuropathic mechanical allodynia to a great extent (*Peirs et al., 2021*). In addition, spinal CCK⁺ neurons also account for the thermal hyperalgesia (*Wang et al., 2022*). However, little is known about how CCK⁺ neurons mediate the nociception. Combined with our results that GPR30 is widely expressed in spinal excitatory interneurons and involved in neuropathic pain modulation, we speculate that CCK⁺ neurons might convey neuropathic hyperalgesia via GPR30. As expected, most CCK⁺ neurons express GPR30, and knockdown of the *Gper1* in CCK⁺ neurons dramatically relieves the pain induced by CCI, thus indicating the vital role of GPR30 in CCK⁺ neurons mediating neuropathic pain. However, it should be noted that half of GPR30⁺ neurons are not co-localized with CCK⁺ neurons, and further studies are needed to explore the function of these GPR30⁺CCK⁻ neurons in neuropathic pain.

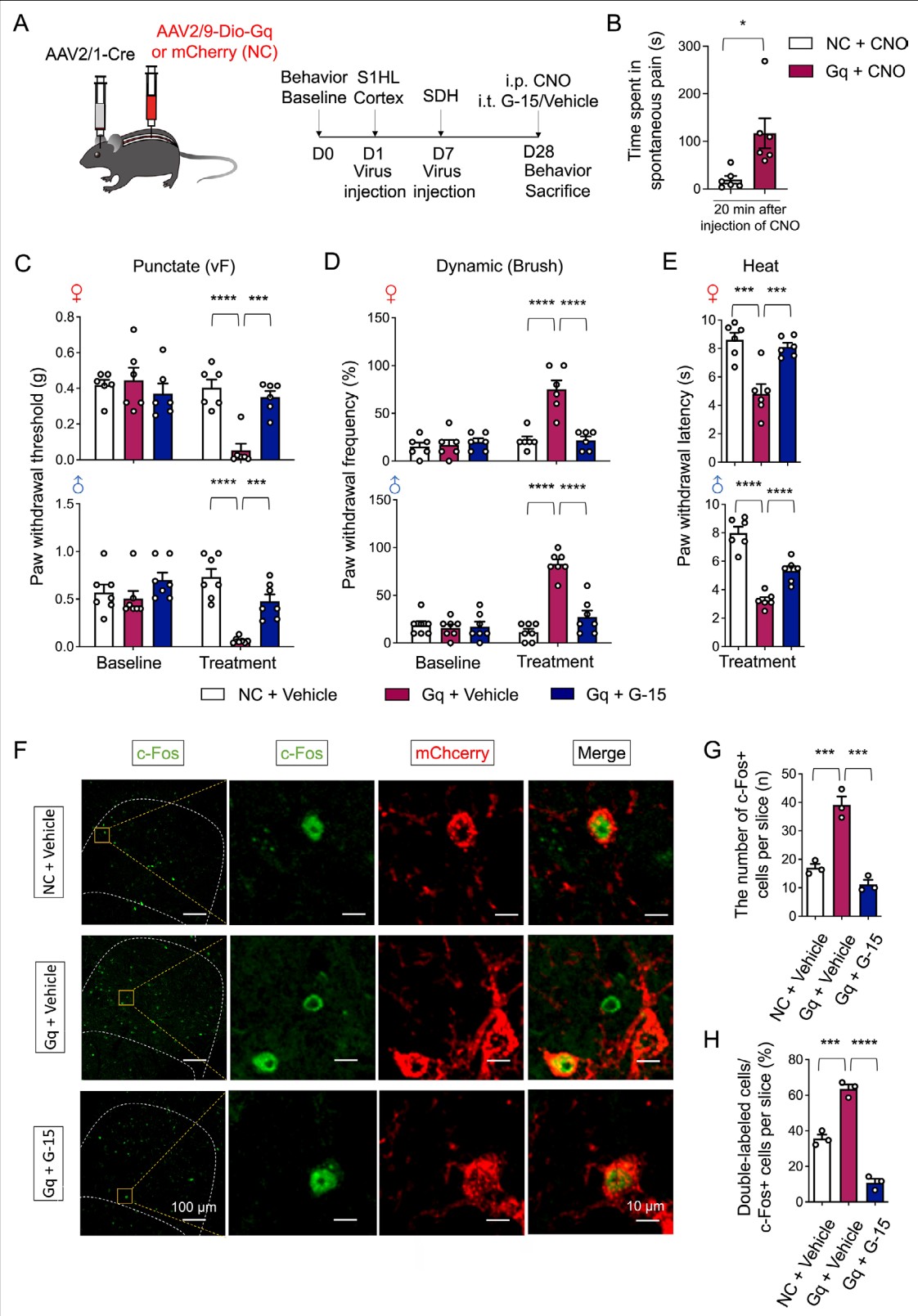

**Figure 6.** Chemogenetic activation of S1-SDH post-synaptic neurons mimicked neuropathic pain symptoms, which were reversed by spinal inhibition of GPR30. (**A**) Schematic illustration of the strategy for identifying the post-synaptic neurons of S1HL-SDH projections in SDH (left) and diagram showing the timeline of AAV2/1 injection in the S1HL cortex, AAV2/9 injection in the spinal dorsal horn, and behavioral tests (right). (**B**) Spontaneous pain induced by intraperitoneal injection of CNO within 20 minutes (n=6 mice for each group). (**C–E**) Behavioral tests of basic nociception, 28 days after brain

*Figure 6 continued on next page*

*Figure 6 continued*

virus injection along with intraperitoneal injection of CNO and intrathecal injection of antagonist of GPR30 or vehicle in Von Frey tests (**C**), Brush tests (**D**), and Heat tests (**E**) in mice of both sexes (n=6 mice for each group). (**F**) Immunochemical detection of c-Fos (green) and m-Cherry+post-synaptic neurons (red). Scale bars: 100 μm. Boxed area of images is enlarged on the right. Scale bars: 10 μm. (**G**) Total number of c-Fos-positive neurons in the SDH per section (n=3 mice for each group, three pictures were analyzed for each mouse). (**H**) Percentage of mCherry positive neurons among c-Fos-positive neurons (n=3 mice for each group, three pictures were analyzed for each mouse). Data information: in (**B**), *p<0.05 (unpaired Student's *t*-test). In (**C, D**), **p<0.01; ***p<0.001; ****p<0.0001 (two-way ANOVA with Turkey's multiple comparisons test). In (**E**), ***p<0.001; ****p<0.0001 (one-way ANOVA with Turkey's multiple comparisons test). In (**G, H**), ***p<0.001; ****p<0.0001. (one-way ANOVA with Turkey's multiple comparisons test). All data are presented as mean ± SEM.

The online version of this article includes the following source data and figure supplement(s) for figure 6:

**Source data 1.** Numerical data and statistic methods used in *Figure 6* and *Figure 6—figure supplement 1*.

**Figure supplement 1.** Chronic constriction injury (CCI)-induced neuropathic pain was relieved by chemogenetic inhibition of S1-SDH post-synaptic neurons.

Abnormal activation of neurons in SDH is one of the causes of hyperalgesia, and the change of post-synaptic currents is the vital factor influencing neuronal excitability (*Zhang et al., 2012b*; *Cao et al., 2022*). Given this critical link between EPSCs and excitability, we measured EPSCs and found an elevation of EPSCs amplitudes in spinal CCK+ neurons after CCI. Furthermore, the knockdown of *Gper1* in CCK+ neurons could inhibit the increase of EPSCs amplitude induced by CCI. Together, these data illustrate that GPR30 promotes the enhancement of synaptic transmission. It should be noted that EPSCs are specifically produced by glutamatergic receptors expressed on post-synaptic membrane, including AMPA and NMDA receptors (*Luo et al., 2016*; *Zhang et al., 2012b*; *Bredt and Nicoll, 2003*). In our study, we confirmed that the selective activation of GPR30 by G-1 remarkably enhanced the AMPA current in spinal CCK+ neurons, which might account for the increased excitability of CCK+ neurons in neuropathic pain. It should be noted that the IPSCs could also influence the excitability of neurons; however, the knockdown of Gper1 failed to change the IPSCs amplitude in CCI mice, suggesting that GPR30 did not take part in the inhibitory synaptic regulation. However, it should be noted that our data did not rule out the potential pre-synaptic contributions.

Increasing evidence has mapped neural circuits from peripheral to central nervous system to illustrate the neural mechanisms of nociception (*Finnerup et al., 2021*; *Cheng et al., 2017*; *Ossipov et al., 2000*; *Peirs and Seal, 2016*; *Arcourt and Lechner, 2015*; *Basbaum et al., 2009*). In brief, pain is derived from the activation of peripheral nociceptors whose cell bodies lie in DRG, and then nociceptive signals are transduced to the SDH for preliminary regulation and finally projected to cerebral cortex via a series of brain region mediating nociception. Additionally, pain is also modulated by the descending modulatory pathways constituted of projections from ventrolateral periaqueductal gray (PAG) to the RVM and then to the SC, which takes several steps (*Porreca et al., 2002*; *Burgess et al., 2002*; *Gardell et al., 2003*; *Bee and Dickenson, 2007*; *Bannister and Dickenson, 2017*; *Lau and Vaughan, 2014*). However, a recent study has come up with the existence of long direct projections from S1 cortex to deep laminae of SDH and the vital role of S1-SDH projections in neuropathic pain (*Liu et al., 2018*). Inhibition of S1-SDH projections attenuates neuropathic pain, while activation decreases pain thresholds in naive mice. We confirmed the existence of these direct projections and their post-synaptic targets' critical role in neuropathic pain. Notably, these long projections specifically synapse within SDH deep laminae onto CCK+ neurons (*Liu et al., 2018*). We also structurally verify that CCK+ neurons receive projections from S1 cortex. Furthermore, we also found that the majority of CCK+ neurons receiving S1-SDH projections express GPR30, thus indicating an important role of GPR30 in descending modulation of S1-SDH. As expected, the knockdown of *Gper1* in S1-SDH post-synaptic neurons dramatically alleviated the hyperalgesia induced by CCI. All these results further suggest an important role of GPR30 in descending facilitation of neuropathic pain. However, since viral strategy only labeled a small fraction of post-synaptic CCK+ neurons from S1 cortex and was insufficient to functionally manipulate these neurons, more efficient methods should be employed to verify the role of GPR30 expressed on S1-SDH post-synaptic CCK+ neurons under neuropathic pain conditions. In addition, manipulation of the S1-SDH projections should be employed in the future to further verify the direct functional connection between corticospinal projections and GPR30.

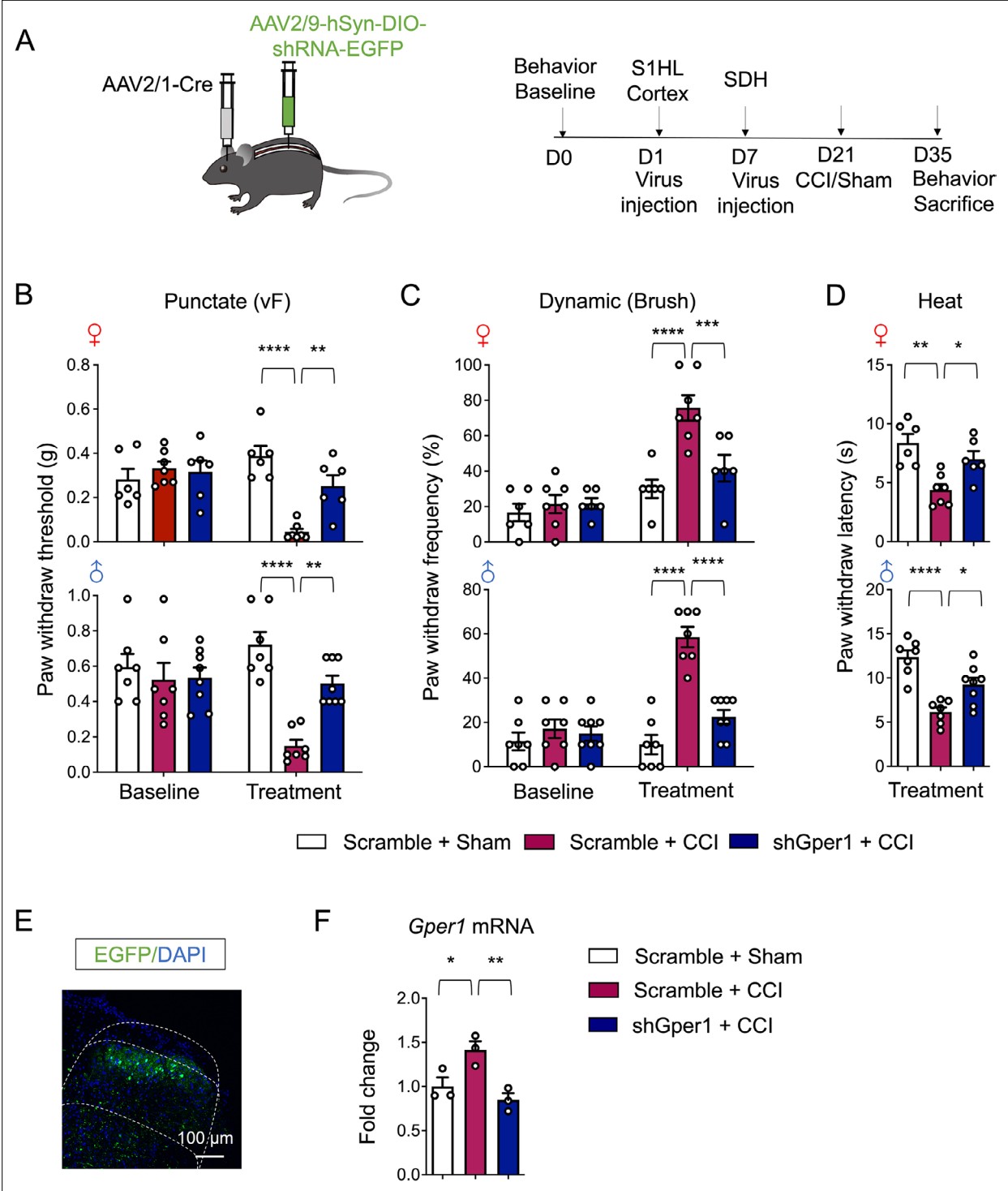

**Figure 7.** Chronic constriction injury (CCI)-induced neuropathic pain was attenuated by knock-down of GPR30 in S1-SDH post-synaptic neurons. (**A**) Schematic illustration of the strategy to knock down Gper1 in the post-synaptic neurons of S1HL-SDH projections in a Cre-dependent manner (left) and the diagram showing the timeline of virus injection, CCI surgery, and behavioral tests (right). (**B–D**) Behavioral tests of basic nociception and 14 days after CCI surgery in Von Frey tests (**B**), Brush tests (**C**), and Heat tests (**D**) in mice of both sexes (n=6 mice for each group). (**E**) Immunochemical detection of the localization of virus expression (green). Scale bars: 100 μm. (**F**) Quantitative PCR analysis of Gper1 mRNA in SDH (n=3 mice for each group). Data information: in (**B, C**), **p<0.01; ***p<0.001; ****p<0.0001 (two-way ANOVA with Turkey's multiple comparisons test). In (**D**), *p<0.05; **p<0.01; ****p<0.0001 (one-way ANOVA with Turkey's multiple comparisons test). In (**F**), *p<0.05; **p<0.01 (one-way ANOVA with Turkey's multiple comparisons test). All data are presented as mean ± SEM.

*Figure 7 continued on next page*

*Figure 7 continued*

The online version of this article includes the following source data for figure 7:

**Source data 1.** Numerical data and statistic methods used in *Figure 7*.

## Conclusion

GPR30 plays a critical role in the development of neuropathic pain, particularly within CCK+ neurons. Our research highlights GPR30's essential function in enhancing AMPA-dependent EPSCs, which are crucial for the activation of CCK$^+$ neurons and the subsequent development of abnormal nociception under neuropathic conditions. Furthermore, GPR30 in post-synaptic neurons of the descending projections via corticospinal projections contributes to the propagation of neuropathic pain signals. These findings suggest that targeting GPR30 in spinal CCK$^+$ neurons could be a promising therapeutic strategy for neuropathic pain in the clinic.

## Methods

### Animals

Mice of both sexes ranging in age from 8 weeks to 12 weeks were used for this study, including C57BL/6J wild-type mice (purchased from SLAC Laboratory Animal CO. LTD, Shanghai, China), *Cck$^{Cre}$* mice (#011086), and *Rosa26$^{tdTomato}$* mice (Ai14, #007941) (originally purchased from Jackson Laboratory). In accordance with the Jackson Laboratory's protocol, transgenic mice were genotyped. All animals were kept in a humidity-controlled room with free access to food and water. The facility was maintained at 22°C and ran on 12 hours of light/dark cycles. A random assignment of animals to different experiment groups was conducted. The animals were treated in accordance with protocols approved by the Animal Ethic and Welfare Committee of Zhejiang University School of Medicine (Permit Number: ZJU20220043), and all experimental procedures were carried out in accordance with the National Institutes of Health Guide for Care and Use of Laboratory Animals (NIH Publications no. 86-23).

### Drug administration

For pharmacological manipulation of the activity of spinal GPR30, G-1 (B5455, APExBIO, USA), or G-15 (diluent of 0.2 mg/mL, administration of 100 µg/kg, 10 µL per mice; B5469, APExBIO, USA) was dissolved in 1% DMSO with saline and administered intrathecally as previously described (*Deliu et al., 2012*). To be specific, mice were lightly anesthetized with 1.5% inhaled isoflurane and held with a pen under the pelvis while a 25-gauge needle attached to a 10 µL syringe (Hamilton, Nevada, USA) was inserted in the subarachnoid space between vertebrae L5 and L6 until a tail flick was observed. The syringe was held for 30 seconds after the injection of 10 µL solution per mice. For chemogenetic manipulation of S1-SDH post-synaptic neurons, Clozapine N-oxide (CNO; 2.5 mg/kg, 150 µL per mice; HY-17366A, MCE, China) was dissolved in saline with gentle vortex for mixing and then administered intraperitoneally (*Wu et al., 2021*). The behavioral assessments were carried out 30 minutes following the injection.

### Virus and CTB microinjection

For intracranial injection, mice were anesthetized with 1% pentobarbital sodium solution (70 mg/kg per mice) and then secured in a stereotaxic frame (RWD Life Science, Shenzhen, China). A middle scalp incision exposed the skull, and then a hole was drilled on the skull above the right S1 cortex to allow passage of a glass microelectrode filled with the virus. Viral injections were performed with the following coordinates of S1: 0.95–1.15 mm from bregma, 1.4–1.6 mm from midline, and 0.9–1.1 mm ventral to skull. A volume of 300 nL virus was injected at 50 nL/min with calibrated glass microelectrodes by a microsyringe pump (#78-8710 KD Scientific, USA). 5 minutes after infusion, the micropipette was slowly removed. For SC injection, mice were anesthetized with 1% pentobarbital sodium solution. The SC could be visible between T12 and T13 vertebral spines following a middle incision along the lumbar vertebrae. With a stereotaxic frame, a glass microelectrode was inserted between L3-L4 SC to a depth of –400 um below the dura, avoiding the posterior spinal arteries. With a stereotaxic injector, 500 nL of viral solution or CTB-555 (1% in phosphate-buffered saline [PBS]; CTB-02,

BrainVTA, Wuhan, China) was slowly infused over a period of 5 minutes. The micropipette was left in place for 5 minutes after infusion before being slowly removed.

To knock down the *Gper1* in CCK$^+$ neurons, AAV2/9-CMV-DIO-(EGFP-U6)-shRNA (GPR30)-WPRE-pA (5×10$^{12}$ v.g./mL) or AAV2/9-CMV-DIO-(EGFP-U6)-shRNA (Scramble)-WPRE-pA (5×10$^{12}$ v.g./mL) was injected into the lumbar SDH of *CCK$^{Cre}$* mice. For visualization of the CCK+ neurons, AAV2/9-CMV-DIO-EGFP-WPRE-pA (5.2×10$^{12}$ v.g./mL) was injected into the lumbar SDH of *CCK$^{Cre}$* mice. For anterograde tracing of S1 cortex projections, AAV2/9-hSyn-EGFP-WPRE (1×10$^{13}$ v.g./mL) was injected into the S1 cortex of wild-type mice. For visualization of the S1-SDH post-synaptic neurons, AAV2/1-hSyn-CRE-WPRE-pA (1×10$^{13}$ v.g./mL) was injected into the S1 cortex and AAV2/9-EF1α-DIO-mCherry-WPRE-pA (1×10$^{13}$ v.g./mL) was injected into the lumbar SDH in wild-type mice. For visualization of CCK$^+$ post-synaptic neurons of S1-SDH projections, AAV2/1- EF1α-FLP-WPRE-pA (1×10$^{13}$ v.g./mL) was injected into the S1 cortex and AAV2/8-hSyn-Con/Fon-EYFP-WPRE-pA (2×10$^{12}$ v.g./mL) was injected into the lumbar SDH of *CCK$^{Cre}$* mice. For chemogenetic manipulation of S1-SDH post-synaptic neurons, AAV2/1-hSyn-CRE-WPRE-pA (1×10$^{13}$ v.g./mL) was injected into the S1 cortex, while AAV2/9-hSyn-DIO-hM3Dq (Gq)-mCherry (3.3×10$^{13}$ v.g./mL; dilution: 1:5) or AAV2/9-hSyn-DIO-hM4Di (Gi)-mCherry (3.3×10$^{13}$ v.g./mL; dilution: 1:5) or AAV2/9-hSyn-DIO-mCherry (3×10$^{13}$ v.g./mL; dilution: 1:5) was injected into the lumbar SDH in wild-type mice. All viruses mentioned above were purchased from BrainVTA. For visualization of the localization of excitatory interneurons in the SDH, AAV2/9-hSyn-mCaMkIIa-mCherry-WPRE-pA (1×10$^{13}$ v.g./mL; Taitool Bioscience, Shanghai, China) was injected into the lumbar SDH of wild-type mice.

For structural tests, at least three mice were examined, and each mouse was examined at least three slices. For behavioral tests, at least six mice per group were examined. The mice with improper position or expression of virus were excluded.

## Chronic constriction injury

The CCI-induced neuropathic pain model was employed as previously documented (*Xia et al., 2021*). Mice were lightly anesthetized via inhaled 1.5% Isoflurane. An incision was made on the skin of each mouse, exposing the sciatic nerve. Four ligations with 6–0 chromic silk were loosely tied around the sciatic nerve. Nerve constriction should be minimal until a brief twitch can be observed. In sham mice, the sciatic nerve was exposed without ligation. The animal was allowed to recover from surgery for 2 weeks before behavioral testing.

## Behavioral test

### Punctate mechanical stimuli (Von Frey filaments)

Mice were habituated to opaque cage (7.5 × 15 × 15 cm) for 1 hour the day before and 30 minutes immediately prior to testing. Testing was performed using a series of Von Frey filaments using the Dixon's up-down method (*Dixon, 1980*), beginning with the 0.16 g filament. The 50% paw withdrawal threshold was determined for each mouse on one hind paw. Each filament was gently applied to the plantar surface of the hind paw for 5 seconds or until a response such as a sharp withdraw, shaking, or licking of the limb was observed. Between individual measurements, filaments were applied at least 3 minutes after the mice had returned to their initial resting state.

### Dynamic mechanical stimuli (brush)

Each mouse was habituated in an opaque cage (7.5 × 15 × 15 cm) for 1 hour the day before and 30 minutes immediately prior to testing. The plantar hind paw was stimulated by light stroking from heel to toe with a paintbrush. A positive response was recorded if the animal was lifting, shaking, or licking the limb. The application was repeated 10 times with a 3-minute interval between each stimulation.

### Plantar heat test (Hargreaves method)

Mice were placed in an acrylic chamber on a glass table and allowed to acclimate to the test chamber for 1 hour the day before and 30 minutes immediately prior to testing. The thermal paw withdrawal latency was assessed using the plantar test (Ugo Basile Biological Research Apparatus, Gemonio, Italy). While the mouse was in a motionless state, a radiant heat source, which was maintained at 40 W, was applied to the plantar surface of the mouse's paw through the glass plate. The paw withdrawal

latency was defined as the time to withdrawal of the hind paw from the heat source, and 15 seconds was used as the cut-off to avoid injury.

## Real-time place escape/avoidance test

Each mouse was habituated in the test room for 1 hour the day before and 30 minutes immediately prior to testing. The RT-PEA chamber (50×28×32 cm; made with plastic plates that had distinct color with another) was placed on the mesh floor. The tested mouse was placed in a two-chamber box and allowed to explore both chambers without any stimulation (pre-stimulation, 10 minutes); mechanical stimulation by 0.16 g von Frey filament was intermittently delivered whenever the mouse entered or stayed in the preferred chamber, as shown in the pre-stimulation stage (stimulation, 10 minutes); the mouse then freely explored the box without any stimulation (post-stimulation, 10 minutes). The mouse's movements and time spent in the preferred chamber were recorded via an ANY-Maze system.

## Immunohistochemistry

Mice were deeply anesthetized with 1% pentobarbital sodium solution and then perfused with PBS followed by pre-cooled 4% paraformaldehyde fix solution (PFA). For c-Fos staining, Von Frey filament with the same force (0.16 g) representing light mechanical stimulation was applied to the right hind paw of each group every 30 seconds for 20 minutes. Then animals were perfused with PBS and 4% PFA 90 minutes after Von Frey filament stimulation. Tissues were harvested and post-fixed in PFA at 4°C overnight before being dehydrated in 30% sucrose for 2 days. Tissues were embedded in Optimal Cutting Temperature (OCT; SAKURA, Japan) and then cut into 10–30 µm sections placed directly onto slides. Tissue slices were blocked at room temperature for an hour with block solution containing 10% normal donkey serum (NDS; 017-000-121, ImmunoResearchLab, USA), 1% bovine serum albumin (BSA; A2153, Sigma, USA) and 0.3% triton X-100 in PBS (PBS-T), and then incubated with primary antibodies diluted in 1% NDS, 1% BSA in PBS-T at 4°C overnight. Sections were washed in PBS and incubated with secondary antibodies at room temperature for 1–2 hours. Slices were washed and covered with Fluoromount-G containing DAPI (0122-20, SouthernBiotech, USA). All images were taken with an Olympus FV1000 confocal microscope. Antibodies used were as follows: anti-c-Fos (1:1000, guinea pig, OB-PGP080-01, Oasis Biofarm, Hangzhou, China), anti-GPR30 (1:500, rabbit, AER-050, Alomone Labs, Israel), anti-IBA1 (1:1000, goat, NB100-1028, Novusbio, USA), anti-GFAP (1:1000, mouse, 3670, Cell Signaling Technology, USA), IB4-FITC (1:1000, I21411, Thermo Fisher, USA), Nissl (1:500, N21483, Thermo Fisher), goat anti-guinea pig IgG-488 (1:500, GP488, Oasis Biofarm), donkey anti-rabbit IgG-488 (1:500, A32790, Thermo Fisher), donkey anti-rabbit IgG-555 (1:500, A32794, Thermo Fisher), donkey anti-mouse IgG-488 (1:500, A21202, Thermo Fisher), and donkey anti-goat IgG-488 (1:500, Ab50129, Abcam, USA).

## In situ hybridization

To verify the specificity of transgenic mice, in situ hybridization was performed according to the manufacturer's instructions from RNAscope Multiplex Fluorescent Reagent Kit v2 (323100, Advanced Cell Diagnostics, USA) with custom-designed probe for *Cck* (Mm-*Cck*-C1, 402278, Advanced Cell Diagnostics). According to the protocols, coronal lumbar spinal sections (10 µm) collected and used for fluorescence in situ hybridization to detect CCK+ neurons. The slice used to stain RFP primary antibody and *Cck* probe was taken from the –80°C refrigerator and immediately incubated with pre-cooled 4% PFA for 15 minutes, followed by gradient dehydration (50% ethanol, 70% ethanol, 100% ethanol, and 100% ethanol, 5 minutes for each gradient). The slides were then incubated with RNAscope hydrogen peroxide at room temperature for 10 minutes, rinsed with distilled water and PBS. Anti-RFP primary antibody (1:1000, rabbit, 600-401-379, Rockland, USA) prepared with co-detection diluent (323180, Advanced Cell Diagnostics) was then added to the slices and incubated overnight at 4°C for subsequent in situ hybridization staining. To characterize the specific expression of GPR30 in SDH, *Slc32a1* (Mm-*Slc32a1*-C1, 319191, Advanced Cell Diagnostics) and *Slc17a6* (Mm-Slc17a6-C3, 319171-C3, Advanced Cell Diagnostics) probe were used as described above as well as anti-GPR30 primary antibody (1:250). At least three mice were examined, and each mouse was examined at least three slices in these experiments.

## Real-time PCR

Mice lumbar SCs or DRG were collected on day 4 after CFA and on day 14 after CCI. Tissues were rapidly collected, frozen in liquid nitrogen, and stored at –80°C. RNA was extracted with standard

procedures using FastPure Cell/Tissue Total RNA Isolation Kit V2 (RC112-01, Vazyme, Nanjing, China). 500 ng of total RNA from each sample was reverse-transcribed with HiScript III RT SuperMix for qPCR (+gDNA wiper) (R312-01, Vazyme). Expression of each mRNA was quantified using ChamQ Universal SYBR qPCR Master Mix (Q711-02, Vazyme). The sequences of quantitative PCR primers were as follows: *Gper1*: F: CCTCTGCTACTCCCTCATCG, R: ACTATGTGGCCTGTCAAGGG; GAPDH: F: AAGAAGGTGGTGAAGCAGGCATC, R: CGGCATCGAAGGTGGAAGATG.

## Spinal slice preparation and whole-cell recording

Spinal slices were prepared as previously described (*Winters et al., 2018*). Mice (6–8-week-old, 2 weeks after CCI surgery) were anesthetized with 1% pentobarbital sodium solution and perfused with ice-cold oxygenated (95% $O_2$ and 5% $CO_2$) cutting artificial cerebrospinal fluid (ACSF, in mM: 100 sucrose, 63 NaCl, 2.5 KCl, 1.2 $NaH_2PO_4$, 1.2 $MgCl_2$, 25 glucose, and 25 $NaHCO_3$), and the SC was rapidly removed. Transverse SC slices (300 µm, L4 to L6 segment) were prepared using a vibratome (VT1200S, Leica, Germany) and incubated in oxygenated NMDG-ACSF (in mM: 93 NMDG, 2.5 KCl, 1.2 $NaH_2PO4$, 30 $NaHCO_3$, 20 HEPES, 25 glucose, 5 Na ascorbate, 2 thiourea, 3 Na pyruvate, 10 $MgSO_4$, and 0.5 $CaCl_2$, and adjusted to pH 7.4 with HCl) at 34°C for 15 minutes. The slices were then transferred to normal ACSF (in mM: 125 NaCl, 2.5 KCl, 1.25 $NaH_2PO_4$, 1.2 $MgCl_2$, 2.5 $CaCl_2$, 25 glucose, and 11 $NaHCO_3$) at 34°C for 1 hour and maintained at room temperature before recording. The slices were transferred to a recording chamber perfused with normal ACSF saturated with 95% $O_2$ and 5% $CO_2$.

Whole-cell patch-clamp recordings were performed using a Heka EPC 10 amplifier (Heka Elektronik). Borosilicate glass pipettes with the resistance of 3–5 MΩ were pulled using a horizontal pipette puller (P97, Sutter Instruments, USA). The pipettes were filled with cesium-based intracellular fluid (in mM: 100 $CsCH_3SO_3$, 20 KCl, 10 HEPES, 4 Mg-ATP, 0.3 Tris-GTP, 7 $Tris_2$-Phosphocreatine, 3 QX-314; pH 7.3, 285–290 mOsm). Targeted whole-cell recordings were made from EGFP-expressing neurons in slices taken from *CCK^Cre* mice with virus injection. For sEPSCs recording, the membrane potential was held at –70 mV, and for spontaneous inhibitory post-synaptic currents (sIPSCs) recording, the membrane potential was held at + 10 mV. Then sEPSCs and sIPSCs were recorded for 100 seconds and analyzed using MiniAnalysis software (Synaptosoft). Five mice per group were examined in the experiment.

To record AMPA-mediated EPSCs, an electrode placed in the deep laminae of SDH was stimulated at every 15 seconds, and the CCK[+] neurons were voltage clamped at –70 mV. Meanwhile, the spinal slices were incubated with ACSF containing APV (100 µM, ab120003, Abcam) to block NMDA receptors and bicuculline (20 µM, ab120108, Abcam) and strychnine (0.5 µM, S450162, Sigma) to block inhibitory synaptic events. AMPAR-mediated EPSCs were recorded for 15 consecutive responses after stable baseline before and after G-1 application (0.1 µM) to compare to the effect of G-1 on AMPAR-mediated EPSCs. Four mice examined in the experiment.

## Statistical analyses

All experiments were randomized. Animals were randomly chosen from multiple cages. For behavior experiments, measurements were taken blinded to condition. All data are reported as mean ± SEM. The required sample sizes were estimated on the basis of our past experience. Statistical analysis was performed using GraphPad Prism V6. Normal distribution was performed using SPSS V20. For all experiments, $p < 0.05$ was considered to be statistically significant.

## Acknowledgements

This work was supported by the National Natural Science Foundation of China grants (82371220, 82171206) and 4+X Clinical Research Project of Women's Hospital, School of Medicine, Zhejiang University (ZDFY2022-4XA102). This work was also supported by the Fundamental Research Funds for the Central Universities (2023ZFJH01-01, 2024ZFJH01-01, and 226-2022-00227). We also thank Sanhua Fang from the Core Facilities, Zhejiang University School of Medicine, for the excellent technical assistance.

# Additional information

## Funding

| Funder | Grant reference number | Author |
| --- | --- | --- |
| National Natural Science Foundation of China | 82371220 | Xinzhong Chen |
| National Natural Science Foundation of China | 82171206 | Zhen-Zhong Xu |
| 4+X Clinical Research Project of Women's Hospital, School of Medicine, Zhejiang University | ZDFY2022-4XA102 | Xinzhong Chen |
| Fundamental Research Funds for the Central Universities | 2023ZFJH01-01 | Zhen-Zhong Xu |
| Fundamental Research Funds for the Central Universities | 2024ZFJH01-01 | Zhen-Zhong Xu |
| Fundamental Research Funds for the Central Universities | 226-2022-00227 | Zhen-Zhong Xu |

The funders had no role in study design, data collection and interpretation, or the decision to submit the work for publication.

## Author contributions

Qing Chen, Conceptualization, Data curation, Formal analysis, Investigation, Methodology, Writing - original draft; Hui Wu, Shulan Xie, Formal analysis, Investigation, Methodology; Fangfang Zhu, Fang Xu, Lihong Sun, Linghua Xie, Jiaqian Xie, Hua Li, Ange Dai, Methodology; Qi Xu, Wenxin Zhang, Luyang Wang, Cuicui Jiao, Investigation; Yue Yang, HongHai Zhang, Visualization; Xuelong Zhou, Writing – review and editing; Zhen-Zhong Xu, Xinzhong Chen, Conceptualization, Data curation, Formal analysis, Supervision, Funding acquisition, Project administration, Writing – review and editing

## Author ORCIDs

Hui Wu ⓘ https://orcid.org/0000-0003-4907-3625
HongHai Zhang ⓘ https://orcid.org/0000-0003-3530-2060
Zhen-Zhong Xu ⓘ https://orcid.org/0000-0001-6578-211X
Xinzhong Chen ⓘ https://orcid.org/0000-0001-9219-1681

## Ethics

The animals were treated in accordance with protocols approved by the Animal Ethic and Welfare Committee of Zhejiang University School of Medicine (Permit Number: ZJU20220043), and all experimental procedures were carried out in accordance with the National Institute of Health Guide for Care and Use of Laboratory Animals (NIH Publications NO.86-23).

Reviewer #1 (Public review): https://doi.org/10.7554/eLife.102874.4.sa1
Reviewer #3 (Public review): https://doi.org/10.7554/eLife.102874.4.sa2
Author response https://doi.org/10.7554/eLife.102874.4.sa3

# Additional files

## Supplementary files

MDAR checklist

**Data availability**

*Figure 1—source data 1*, *Figure 2—source data 1*, *Figure 3—source data 1*, *Figure 4—source data 1*, *Figure 5—source data 1*, *Figure 6—source data 1*, *Figure 7—source data 1* contain the numerical data used to generate the figures.

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
