## [Editor Report · eLife Assessment]

This **important** study investigates nerve-injury-induced allodynia by studying the role of a subpopulation of excitatory dorsal horn CCK+ neurons that express the estrogen receptor GPR30 and potentially modulate nociceptive sensitivity via direct inputs from primary somatosensory cortex. In this revised version, the authors addressed many of the critiques raised through added analyses that **convincingly** support the notion that spinal GPR30 neurons are indeed an excitatory subpopulation of CCK+ neurons that contribute to neuropathic pain. While evidence of a direct functional corticospinal projection to CCK+/GPR30+ neurons is not fully demonstrated, this work will be of broad interest to researchers interested in the neural circuitry of pain.

---

## [Referee Report · Reviewer #1 (Public review)]

In this manuscript, Chen et al. investigate the role of the membrane estrogen receptor GPR30 in spinal mechanisms of neuropathic pain. Using a wide variety of techniques, they first provide convincing evidence that GPR30 expression is restricted to neurons within the spinal cord, and that GPR30 neurons are well-positioned to receive descending input from the primary sensory cortex (S1). In addition, the authors put their findings in the context the previous knowledge in the field, presenting evidence demonstrating that GRP30 is expressed in the majority of CCK-expressing spinal neurons. Overall, this manuscript furthers our understanding of neural circuity that underlies neuropathic pain and will be of broad interest to neuroscientists, especially those interested in somatosensation. Nevertheless, the manuscript would be strengthened by additional analyses and clarification of data that is currently presented.

Strengths:

The authors present convincing evidence for expression of GPR30 in the spinal cord that is specific to spinal neurons. Similarly, complementary approaches including pharmacological inhibition and knockdown of GPR30 are used to demonstrate a role for the receptor in driving nerve injury-induced pain in rodent models.

Weaknesses:

Although steps were taken to put their data into the broader context of what is already known about the spinal circuitry of pain, more considerations and analyses would help the authors better achieve their goal. For instance, to determine whether GPR30 is expressed in excitatory or inhibitory neurons, more selective markers for these subtypes should be used over CamK2. Moreover, quantitative analysis of the extent of overlap between GPR30+ and CCK+ spinal neurons is needed to understand the potential heterogeneity of the GPR30 spinal neuron population, and to interpret experiments characterizing descending SI inputs onto GPR30 and CCK spinal neurons. Filling these gaps in knowledge would make their findings more solid.

Revised Manuscript Update:

In their revised manuscript, Chen et al. have added additional data that establishes GPR30 spinal neurons as a population of excitatory neurons, half of which express CCK. These data help to position GPR30 neurons in the existing framework of spinal neuron populations that contribute to neuropathic pain, strengthening the author's findings.

I have no new recommendations to the author's following this round of revisions.

---

## [Referee Report · Reviewer #3 (Public review)]

Summary:

The authors convincingly demonstrate that a population of CCK+ spinal neurons in the deep dorsal horn express the G protein coupled estrogen receptor GPR30 to modulate pain sensitivity in the chronic constriction injury (CCI) model of neuropathic pain in mice. Using complementary pharmacological and genetic knockdown experiments they convincingly show that GPR30 inhibition or knockdown reverses mechanical, tactile and thermal hypersensitivity, conditioned place aversion, and c-fos staining in the spinal dorsal horn after CCI. They propose that GPR30 mediates an increase in postsynaptic AMPA receptors after CCI using slice electrophysiology which may underlie the increased behavioral sensitivity. They then use anterograde tracing approaches to show that CCK and GPR30 positive neurons in the deep dorsal horn may receive direct connections from primary somatosensory cortex. Chemogenetic activation of these dorsal horn neurons proposed to be connected to S1 increased nociceptive sensitivity in a GPR30 dependent manner. Overall, the data are very convincing and the experiments are well conducted and adequately controlled. The potential role of direct connections from S1 for descending modulation of pain and the endogenous mechanism(s) activating GPR30 will be interesting to test in future studies.

Strengths:

The experiments are very well executed and adequately controlled throughout the manuscript. The data are nicely presented and supportive of a role for GPR30 signaling in the spinal dorsal horn influencing nociceptive sensitivity following CCI. The authors also did an excellent job of using complementary approaches to rigorously test their hypothesis.

Weaknesses:

While the viral tracing demonstrates a potential connection between S1 and CCK+ or GPR30+ spinal neurons, no direct evidence is provided for S1 in facilitating any activity of these neurons in the dorsal horn.

Comments on the latest version:

The authors have done a good job addressing previous critiques and have appropriately revised the manuscript and conclusions.

---

## [Author Response]

The following is the authors’ response to the previous reviews

**Reviewer #1 (Public review):**
In their revised manuscript, Chen et al. have added additional data that establishes GPR30 spinal neurons as a population of excitatory neurons, half of which express CCK. These data help to position GPR30 neurons in the existing framework of spinal neuron populations that contribute to neuropathic pain, strengthening the author's findings.

Thank you very much for your positive feedback and for recognizing the value of our additional data.

**Reviewer #3 (Public review):**
The authors did an excellent job addressing many of the critiques raised. Despite acknowledging that a direct functional corticospinal projection to CCK/GPR30+neurons is not supported by the data and revising the title, these claims still persist throughout the manuscript. Manipulating gene expression or the activity of postsynaptic neurons through a trans-synaptic labeling strategy does not directly support any claim that those upstream neurons are directly modulating spinal neurons through the proposed pathway. Indeed they might, but that is not demonstrated here.

We sincerely thank the reviewer for this critical insight. We fully agree that our trans-synaptic approach does not provide a direct functional connection. In response, we have revised the manuscript to remove any overstated claims of "direct" modulation and instead emphasize the critical role of spinal GPR30+ neurons. Moreover, we have added a statement in the Discussion to acknowledge this limitation and to highlight that the precise function role of this connection requires further investigation in further studies.

**Reviewer #1 (Recommendations for the authors):**
I recommend 2 minor corrections to the text and figures(1) Line 131 : "What's more, near-universal CCK+ neurons were co-localized with GPR30 (Fig 2F and G)."The additional quantification of the overlap between GPR30 and tdTomato provided by the authors is useful, but there are inconsistencies with how the data are reported in the figures and text, making them difficult to interpret. 2F supports the author's conclusion that approximately 90% of CCK⁺ neurons express GPR30, and about 50% of GPR30⁺ neurons co-express CCK. However, the x-axis labels in 2G appear to have been switched, and suggest that the opposite is true (i.e., most GRPR neurons are CCK+, while only 50% of CCK neurons are GPR30+). Please clarify which is correct throughout the results and discussion sections.

Thank you for identifying this important error. We apologized for the confusion caused by the mislabeled x-axis in Fig. 2G. The x-axis labels were indeed inadvertently switched. The correct data is that approximately 90% of CCK^+^ neurons express GPR30. We have corrected the figure and have carefully reviewed the entire manuscript to ensure all related descriptions and discussions are consistent with the accurate quantification.

(2) The following sentence describing Figure 5 was hard to follow: Lines 190-192, "Consistent with prior observations, we found that these SDH downstream neurons exhibited colocalization with CCK+ neurons, with 28.1% of mCherry+ neurons expressing CCK (Fig 5I and J)." Since the authors are describing a common population of neurons, a statement describing the coexpression (rather than the colocalization" would more simply summarize their data.)

We thank the reviewer for this helpful suggestion. We fully agree that "coexpression" is a more precise term for the description. We have revised the sentence on Lines 189-190 to read: "Consistent with prior observations, we found that 28.1% of mCherry+ S1-SDH downstream neurons coexpressed CCK (Fig 5I and J)."

**Reviewer #3 (Recommendations for the authors):**
Additional RecommendationsThe authors did a commendable job revising the manuscript text to improve readability; however, several informal phrases from the original version still persist, or were added (e.g. "by the way").

We thank the reviewer for this valuable feedback regarding the language. We have conducted a line-by-line review of the entire manuscript to identify all remaining informal phrases, and replaced them with more appropriate phrasing.

It should be clearly mentioned that spontaneous E/IPSCs were recorded in Figure 4 and Fig S5.

We thank the reviewer for this helpful suggestion. We have now clearly indicated the spontaneous E/IPSCs in Fig. 4 and Fig. S5 and manuscript.

The rationale for recording EPSCs from GFP-labeled CCK+ neurons because "a significant proportion of spinal CCK+ neurons form excitatory synapses with upstream neurons" does not make any sense. Do the authors instead mean that CCK neurons receive excitatory inputs from other spinal neurons and intend to test if those synaptic connections are modulated by GPR30?

We thank the reviewer for this critical correction. Our intended meaning was indeed that CCK^+^ neurons receive excitatory inputs from other neurons, and we aimed to test whether those synaptic connections are modulated by GPR30. To avoid confusion, we have revised the manuscript to remove the erroneous statement “Since CCK+ neurons mainly receive excitatory synaptic inputs from upstream neurons, we then intended to test whether GPR30 modulated these synaptic connections.”

I am confused by the statement on Page 8 "to examine whether GPCR30-mediated EPSCs depend on AMPA mediated currents." Given that sEPSCs were recorded at -70 mV in low Cl internal I'm not sure what other glutamate receptor would be involved. Perhaps the intention was to more directly test whether GPR30 activation acutely modulates AMPAR-mediated EPSCs? However, as the authors acknowledged, this experiment does not necessarily support a solely post-synaptic AMPAR-dependent mechanism.

We thank the reviewer for this insightful comment and apologize for the lack of clarity. Our intention was indeed to test whether GPR30 activation modulates AMPAR-mediated currents. We have revised the text. In addition, we also emphasize in the Discussion that our data did not rule out the potential pre-synaptic contributions to this effect.

An elevation in EPSCs within a cell does not necessarily mean that the cell is more excitable, only that it is receiving more excitatory inputs or has an increase in synaptic receptors. The cell may scale down its activity to compensate for this increase. I recommend only drawing conclusions from what the experiments actually tested.

We thank the reviewer for this crucial clarification. We have revised the manuscript to remove any claims that the cells were "more excitable". Our conclusions now strictly focus on the specific findings that GPR30 activation enhanced the excitatory transmission onto CCK^+^ neurons.